# Cell-cycle-dependent repression of histone gene transcription by histone H4

Kami Ahmad [1] ✉, Matt Wooten[1], Brittany N. Takushi[1], Velinda Vidaurre[2], Xin Chen [2,3] & Steven Henikoff [1,3] ✉

In all eukaryotes, DNA replication is coupled to histone synthesis to coordinate chromatin packaging of the genome. Canonical histone genes coalesce in the nucleus into the histone locus body (HLB), where gene transcription and 3′ mRNA processing occurs. Both histone gene transcription and mRNA stability are reduced when DNA replication is inhibited, implying that the HLB senses the rate of DNA synthesis. In *Drosophila melanogaster*, the S-phase-induced histone genes are tandemly repeated in an ~100 copy array, whereas, in humans, these histone genes are scattered. In both organisms, these genes coalesce into HLBs. Here, we use a transgenic histone gene reporter and RNA interference in *Drosophila* to identify canonical H4 histone as a unique repressor of histone synthesis during the G2 phase in germline cells. Using cytology and CUT&Tag chromatin profiling, we find that histone H4 uniquely occupies histone gene promoters in both *Drosophila* and human cells. Our results suggest that repression of histone genes by soluble histone H4 is a conserved mechanism that coordinates DNA replication with histone synthesis in proliferating cells.

The genome of eukaryotic cells is packaged into nucleosomes, where DNA is wrapped around histone octamers. In animal cells, the genes encoding canonical histone proteins are highly distinctive. These multicopy genes are abundantly transcribed by RNA polymerase II (RNAPII) during the S phase of the cell cycle and are the only protein-coding genes that produce transcripts without introns or 3′ polyadenylation[1]. Histone genes nucleate a distinctive body within the nucleus termed the histone locus body (HLB), where specific transcription factors and RNA processing proteins localize. The HLBs of *Drosophila* and mammals share fundamental molecular components, including the cyclin E/CDK2-activated transcription cofactor Mxc/NPAT, 3′ mRNA stem loops, stem-loop-binding protein (SLBP) and the U7 small nuclear ribonucleoprotein 3′-end processing machinery. Despite these cytological and compositional similarities, the histone genes are radically different in gene organization. In *Drosophila melanogaster*, the five canonical histone genes (H1, H2A, H2B, H3 and H4) are arranged in a unit tandemly repeated around 100 times at one locus[2], whereas, in humans, 72 genes are scattered with a major cluster on chromosome 6 and two minor clusters on chromosome 1. These human genes are nonrepetitive

and embedded in euchromatic regions of chromosomes, whereas the tandemly repeated *Drosophila* genes are subject to heterochromatic silencing[3] and it has been unclear how many of these 200 gene repeat units in a diploid cell are transcribed. By profiling both *D. melanogaster* and human histone genes, we aim to understand ancient conserved mechanisms of S-phase-induced histone gene regulation that have endured despite profound genomic and epigenomic changes.

Here, we took advantage of *Drosophila* histone gene rescue[4,5] and fluorescently marked histone transgene reporters[6] to test for histone gene derepression after knockdown of candidate repressors in the synchronized G2-phase gonial cells of testes. We discovered that a reduction in histone H4 strongly derepressed histone gene expression outside of S phase but no other candidate regulator had an effect. Using imaging, we showed that histone H4 localizes to the HLB in *Drosophila* cells; using CUT&Tag chromatin profiling, we precisely localized histone H4 to histone gene promoters, coinciding with peaks of Mxc/NPAT, initiating RNAPII and active chromatin. Turning to human K562 cells, we observed similar cytological localization of histone H4 to the HLBs and coincident genomic localization of histone H4, NPAT and RNAPII

[1]Basic Sciences Division, Fred Hutchinson Cancer Center, Seattle, WA, USA. [2]Department of Biology, The Johns Hopkins University, Baltimore, MD, USA. [3]Present address: Howard Hughes Medical Institute, Chevy Chase, MD, USA. ✉e-mail: kahmad@fredhutch.org; steveh@fredhutch.org

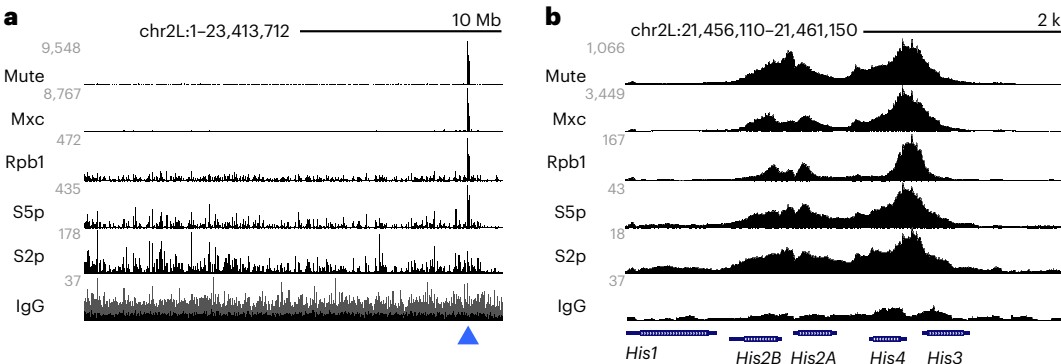

**Fig. 1 | Chromatin factors and RNAPII at the histone locus in wing imaginal disc cells. a**, Browser tracks of chromatin factors and RNAPII isoforms across chromosome 2L of *Drosophila*. The blue arrowhead marks the location of the histone locus. **b**, Browser tracks of chromatin factors and RNAPII isoforms across one HRU.

at active histone genes. These results indicate a direct mechanism whereby excess histone H4 in cells buffers histone gene transcription to coordinate chromatin packaging with DNA replication.

## Results

### Chromatin features at active and silenced histone genes in *Drosophila*

As the histone genes in the HLB are repetitive and normal cells contain both active and silenced histone genes, mapping of chromatin features to a genome assembly cannot distinguish which features are associated with which expression state. To address this, we profiled chromatin features in two genotypes: wild type, where some of the ~200 histone genes must be active while others are silenced, and the '12XWT' line, where the histone locus has been deleted and a construct carrying 12 copies of the histone repeat unit (HRU) rescues the flies[7]. We expect all copies of the histone genes to be active in this second genotype. By comparing chromatin profiles between these two genotypes, we could infer the chromatin features of active and of silenced histone genes.

We dissected wing imaginal discs from male larvae as a sample of proliferating cells and subjected them to CUT&Tag profiling[8], generating >1 million reads mapped to the dm6 genome assembly for each sample (Supplementary Table 1). We first used antibodies to the HLB-specific transcription cofactors Mxc[9] and Mute[10]. As expected, these two factors are localized to only the histone locus in the genome (Fig. 1a). Signal for both factors was broadly dispersed across the *His3*–*His4* and *His2A*–*His2B* gene pairs of the 5-kb HRU, with peaks at the divergent promoters of each pair and no signal over the adjacent *His1* gene (Fig. 1b). These results are consistent with previous chromatin mapping of Mxc in *Drosophila* embryos[11]. The divergent *His4*–*His3* promoter region nucleates HLB formation[12] and these binding profiles support the idea that Mxc binds at these sites in the histone locus and nucleates HLB formation. Furthermore, the lack of a Mute or Mxc signal over the *His1* gene is consistent with the idea that the linker histone gene is controlled by the CRAMP/CRAMP1 transcription factor[13].

To assess transcriptional activity of the histone genes, we profiled multiple components and isoforms of RNAPII in larval wing disc samples. As expected, the RNAPII component unphosphorylated Rpb1 and the phosphorylated initiating (RNAPII-S5p) and elongating (RNAPII-S2p) isoforms of Rpb1 marked the histone locus in wild-type cells (Fig. 1). In fact, the histone locus was the major site of Rpb1 and RNAPII-S5p signal across the genome (Fig. 1a), accounting for 1.8% and 2.5% of mapped reads, respectively, while only 0.2% of reads for the RNAPII-S2p isoform mapped to this locus. This is consistent with cytological description of enrichment for the unmodified and initiating forms of RNAPII at the HLB in other *Drosophila* cell types[14–16]. Note that the *His3*–*His4* and *His2A*–*His2B* pairs had strong peaks of RNAPII-S5p and RNAPII-S2p isoforms at their promoters and across their gene

bodies, indicating high transcription of these genes. By contrast, *His1* had only a low broad distribution of these polymerase isoforms across its length (Fig. 1b), implying that it was expressed at lower levels.

We then compared signal counts for Mxc, Mute and RNAPII between wing disc samples from wild-type and from 12XWT flies. We calculated per-gene counts to adjust for the different numbers of copies in the two strains (100 copies per genome in wild type, 12 copies per genome in 12XWT). The per-copy sum counts of Mute and Mxc signal were higher in wild-type than in 12X flies, and proportional to the numbers of histone genes in these genotypes (Fig. 2a). By contrast, signals for the RNAPII-S5p isoform were dramatically increased and signal for the RNAPII-S2p isoform showed a slight gain. Thus, each histone gene in the 12XWT line carried more RNAPII than in the wild type. There was no significant change in genome-wide gene expression (Extended Data Fig. 1 and Supplementary Table 3), consistent with the observation that the 12XWT rescue construct is sufficient to support viability and fertility[7]. The effect on chromatin features at histone genes implies that, in the wild type, some histone genes were active and others were silenced; alternatively, each gene was active at an intermediate level. In either case, all histone genes in the 12XWT strain appeared to be more heavily transcribed, presumably to support cell proliferation.

### Heterochromatic histone marks at silenced HLB genes

To determine histone modifications associated with histone genes, we first profiled five modifications associated with active gene expression[17] (Fig. 2b). In wild-type samples, all three methylation states of the histone H3K4 residue (H3K4me1, H3K4me2 and H3K4me3) and acetylation at histone H3K27 (H3K27ac) were enriched at the wild-type histone locus at levels comparable to surrounding active enhancers and genes. By contrast, there was little detectable trimethylation of the K36 residue of histone H3 (H3K36me3), consistent with the intronless structure of histone genes[18]. Adjusting for copy number, there was a substantial per-gene copy gain only in the H3K4me3 mark in the 12XWT line (Fig. 2a) and more moderate gains in the H3K4me1 and H3K4me2 marks. These changes are consistent with the higher average transcription of histone genes in this genotype.

We then examined five histone modifications typically associated with silencing across histone genes between the two *Drosophila* lines. Three modifications were strongly enriched at the histone genes in wild-type samples: monomethylation and dimethylation of histone H3 at K9 (H3K9me1 and H3K9me2)[19] and ubiquitinylation of histone H2A at K118 (uH2A)[20] (Fig. 2b). The presence of these first two marks suggests that histone genes have a partially heterochromatic character, likely because of the repeated genes, although the locus lacks trimethylation of histone H3 at K9 (H3K9me3)[19] (Fig. 2b). The presence of uH2A at histone genes suggested they are sites of Polycomb repressive complex 1

**a**

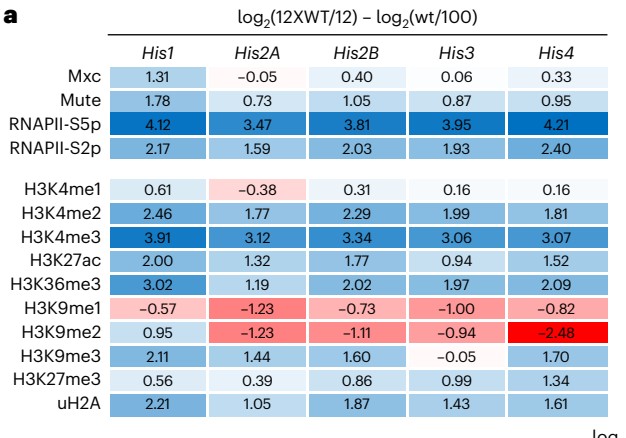

| | | | $\log_2(12\mathrm{XWT}/12) - \log_2(\mathrm{wt}/100)$ | | |
| --- | --- | --- | --- | --- | --- |
| | *His1* | *His2A* | *His2B* | *His3* | *His4* |
| Mxc | 1.31 | −0.05 | 0.40 | 0.06 | 0.33 |
| Mute | 1.78 | 0.73 | 1.05 | 0.87 | 0.95 |
| RNAPII-S5p | 4.12 | 3.47 | 3.81 | 3.95 | 4.21 |
| RNAPII-S2p | 2.17 | 1.59 | 2.03 | 1.93 | 2.40 |
| H3K4me1 | 0.61 | −0.38 | 0.31 | 0.16 | 0.16 |
| H3K4me2 | 2.46 | 1.77 | 2.29 | 1.99 | 1.81 |
| H3K4me3 | 3.91 | 3.12 | 3.34 | 3.06 | 3.07 |
| H3K27ac | 2.00 | 1.32 | 1.77 | 0.94 | 1.52 |
| H3K36me3 | 3.02 | 1.19 | 2.02 | 1.97 | 2.09 |
| H3K9me1 | −0.57 | −1.23 | −0.73 | −1.00 | −0.82 |
| H3K9me2 | 0.95 | −1.23 | −1.11 | −0.94 | −2.48 |
| H3K9me3 | 2.11 | 1.44 | 1.60 | −0.05 | 1.70 |
| H3K27me3 | 0.56 | 0.39 | 0.86 | 0.99 | 1.34 |
| uH2A | 2.21 | 1.05 | 1.87 | 1.43 | 1.61 |

$\log_2\mathrm{FC}$

−2.5 0 +4.2

**b**

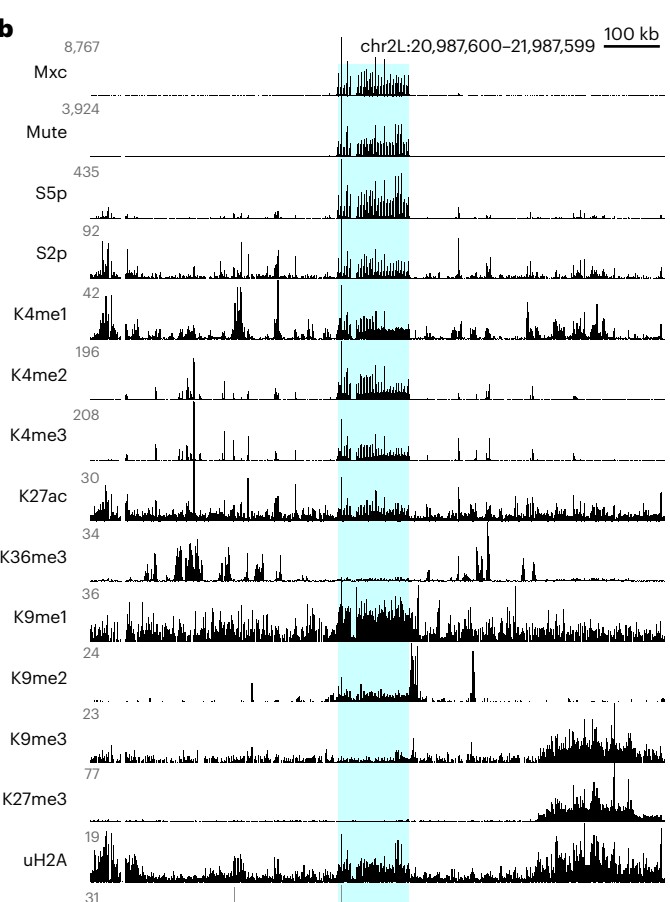

chr2L:20,987,600–21,987,599 100 kb

8,767 — Mxc
3,924 — Mute
435 — S5p
92 — S2p
42 — K4me1
196 — K4me2
208 — K4me3
30 — K27ac
34 — K36me3
36 — K9me1
24 — K9me2
23 — K9me3
77 — K27me3
19 — uH2A
31 — IgG

**Fig. 2 | Histone modifications at the histone locus in wing imaginal disc cells.**
**a**, Per-gene fold change in chromatin features and histone modifications
between the 12XWT strain (12 histone gene copies per genome) and wild type
(100 copies per genome). FC, fold change; wt, wild type. The H3K36me3,
H3K9me3 and H3K27me3 modifications are absent from the wild-type and
12XWT histone genes and these ratios are because of minor changes in signal.
**b**, Browser tracks of histone modifications around the histone locus in wild-type
wing imaginal disc cells. Blue shading marks the histone locus.

(PRC1) activity, although the canonical trimethylation mark of histone
H3 at K27 (H3K27me3)[21] of Polycomb-silenced domains was absent.

As any histone modification associated with histone gene silenc-
ing should be present in the wild type but absent in the 12XWT line,

we calculated per-gene copy changes for the three modifications
that were above background levels across the histone locus (Fig. 2a).
Previous results have implicated H3K9 methylation in histone gene
silencing[22,23]. Indeed, the per-gene copy coverage of the H3K9me1
and H3K9me2 marks dropped in the 12XWT strain compared to the
wild type. By contrast, the per-gene density of the uH2A modification
was slightly increased. These results support the idea that inactive
histone genes are marked with monomethylation and dimethylation
of the H3K9 residue, where the wild-type histone locus is a mixture of
transcriptionally active histone genes mixed with silenced histone
genes. This may be analogous to the functional organization of ribo-
somal RNA genes, where actively transcribed units are intermixed
with silenced units[24]. Alternatively, individual histone genes may
carry both active and repressive modifications that quantitatively
adjust gene expression. In either case, a partially active set of histone
genes may allow cells to fine-tune histone production to the needs of
cell proliferation and growth, the rates of which vary across tissues
and life stages.

### A visual reporter for histone gene silencing

Extra HRU transgenes are repressed in proportion to the number of
total histone genes in a genotype[25], an effect that appears similar to
the reduced expression of histone genes in the wild type. To visualize
histone gene expression in living animals, we used HRU reporter con-
structs where either the *His3* gene or the *His2A* gene was fused to the
octocoral *Dendra2* fluorescent-protein-coding sequence[6]. These con-
structs express fluorescently tagged histones in eggs and developing
embryos[6] and at low levels in proliferating cells of later stages such as in
larval imaginal wing discs (Fig. 3a). We reasoned that, if these transgenes
are partially repressed, then genetically interfering with histone gene
silencing would produce more fluorescent protein. Indeed, the expres-
sion of the *His2ADendra2* HRU transgene was dramatically increased
~37-fold in the 12XWT background (12 HRU copies per genome) com-
pared to its expression in the wild type wing imaginal discs (Fig. 3b). This
implies that cells with reduced histone gene numbers sense a dearth of
histones and specifically upregulate the histone locus to compensate
and provide for chromatin duplication.

We wished to identify the mechanism by which histone genes
are repressed; however, genetic reduction of such a factor might
inhibit viability in the growing wing imaginal disc. Therefore, we

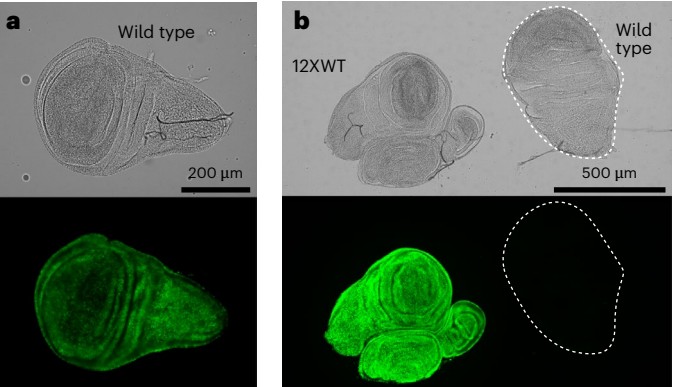

**Fig. 3 | Expression of a *His2ADendra2* reporter HRU in wild-type and 12XWT
strains. a**, Fluorescence of the *His2ADendra2* reporter in a wild-type larval
wing imaginal disc, adjusted to display the weak fluorescence expressed from
the reporter in this genotype. **b**, Fluorescence of *His2ADendra2* in larval wing
imaginal discs from wild-type (100 HRU copies per genome) and 12XWT (12
HRU copies per genome) larvae. The total fluorescence of 12XWT wing imaginal
discs is 37-fold that in the wild-type background (average summed fluorescence
of wild-type discs = 5,908,005 ± 1,096,058 a.u. versus 158,704 + 26,029 a.u. in
the 12XWT background). Imaging of at least five larvae for each genotype was
performed with similar results.

turned to a nonessential tissue where we could still assess reporter expression with RNA interference (RNAi). The adult testis is dispensable for organismal viability. It is also spatially organized such that the developmental and cell-cycle stage of cells can be identified by their position in the tissue[26]. Germline stem cells are located at the apical testis tip and undergo four rounds of mitotic division in this proliferating zone, producing G2-phase spermatogonial cells. These then grow for ~4 days before meiosis and sperm differentiation. We imaged these stages by dissecting and fixing testes from males carrying a *bamGAL4* driver with an inducible *UAS–RFP* transgene. This combination produced red fluorescent protein (RFP) specifically in gonial cells (Fig. 4a). We immunostained these testes with antibodies to Mxc, a constitutive component of the HLB, and to phospho-Mxc/MPM2, which is catalyzed by cyclin E/CDK2 in the S phase of the cell cycle[27]. The proliferating zone at the apical tip of the testis was marked with large HLBs containing phosphorylated Mxc and abutted RFP-stained G2-phase cells (Fig. 4a). Slightly more distal in the testis, the Mxc-labeled HLB was divided into 2–4 smaller dots, consistent with the unpairing of homologous loci in this developmental stage[28]; furthermore, Mxc staining disappeared from nuclei in later primary spermatocytes before meiosis.

We first imaged fluorescence from a control *His2AVDendra2* histone variant gene (located on chromosome 3 outside of the histone locus) and this variant protein was abundant throughout the apical tip of the testis (Fig. 4b,c). By contrast, the HRU transgenes were repressed in the testis. Little or no fluorescence from the *His2ADendra2* transgene was apparent in the very apical tip including in *bam*-positive gonial cells but then weakly appeared in later stages (Fig. 4d,e). Fluorescent protein from the *His3Dendra2* HRU was undetectable throughout the testis, with signal only in the nuclei of somatic sheath cells (Fig. 4f,g). We attribute the difference in pattern of expression between *His2ADendra2* and *His3Dendra2* transgenes to the deposition of histone H2A in postmitotic cells, where histone H3 does not deposit[29,30]. This repression was sensitive to the demand for histones because animals with reduced numbers of histone genes and the *His2ADendra2* (Fig. 4h,i) or *His3Dendra2* (Fig. 4j,k) reporter HRUs showed intense fluorescence throughout the apical tip of the testis, including in gonial cells. A reduction in histone genes did not have a substantial effect on expression of the *His2AVDendra2* reporter (Fig. 4l), indicating that this effect was specific to S-phase-induced histone genes. Thus, we infer that germline cells—like somatic cells—upregulate histone gene expression when these genes are limiting.

## Reduced histone H4 activates histone gene reporters

Previous work has implicated the H3K9 methyltransferase *Suppressor of variegation 3-9* (*Su(var)3-9*) in histone gene silencing[22,23]; indeed, we found that the histone locus was enriched for H3K9 methylation (Fig. 2b). We constructed viable null *Su(var)3-9* flies from *trans*-heterozygous point mutant alleles[31]; however, neither *His2ADendra2* nor *His3Dendra2* reporters were derepressed in this background (Fig. 4m). To identify mechanisms responsible for HRU repression, we targeted a selection of chromatin proteins for RNAi knockdown in the gonial cell stage. None of these knockdowns derepressed *His2ADendra2* or *His3Dendra2* reporters, including the histone H3K9 methyltransferase *eggless* (*egg*)[32], the H3K9me2/3-binding proteins HP1 (ref. 33) or HP2 (ref. 34), the histone H2A ubiquitin ligase *Sex combs extra* (*Sce*)[35], the PRC1 component *Polycomb* (*Pc*)[36] or the histone H3K27 methyltransferase *Enhancer of zeste* (*E(z)*)[37]. We did not assess the effectiveness of these knockdowns in the testis; thus, we cannot rule out that more complete elimination of these histone modifications might affect expression of histone gene reporters.

As changes in histone gene number alter reporter expression, we used RNAi to knock down histones in gonial cells of the male germline. We tested knockdown of the linker histone gene *His1*, the core histone genes *His2A*, *His2B*, *His3* and *His4*, the histone variant genes *His2AV*, *His3.3A* and *His3.3B* and the orphan gene *His4R*. We tested histone knockdown constructs with two available histone GFP reporters and confirmed that these were effective in the testis (Extended Data Fig. 2) but noted that these males contained sperm and were fertile, implying that these knockdowns only partially reduced histone gene expression. The HRU reporters remained repressed in eight of nine histone knockdowns; however, knockdown of the *His4* gene resulted in a ~5-fold increase in *His2ADendra2* expression and ~3-fold increase in *His3Dendra2* expression in gonial cells (Fig. 4m–q). Expression from a variant histone *His2AVDendra2* line was not affected by *His4* knockdown (Fig. 4l). These results suggest that cells measure the demand for S-phase-induced histones only on the basis of the H4 histone.

## Histone H4 localizes to the HLB in *Drosophila* cells

It is surprising that knockdown of only one histone modulated HRU silencing, as histones associate in dimers and in octamers as nucleosomes are assembled. However, some examples have been identified where monomeric histones exist[38] and where a singular histone is used to measure chromatin in both *Drosophila* and in human cells[39,40]. To test whether histone H4 localizes to the HLB on its own,

---

**Fig. 4 | Silencing and derepression of extra histone genes in the *Drosophila* male germline. a**, The HLB in testis cells is marked by Mxc staining (blue). Cells in the proliferating zone at the apical tip of the testis are marked with phospho-Mxc detected by the MPM2 antibody at the HLB (green). Postmitotic G2-phase germline cells are marked by *bamGAL4*-induced *UAS–RFP* expression (red), while the decondensed nuclei of later stages have low DAPI staining (gray). The proliferating zone of testes is marked by the green bar and the RFP-marked G2 gonial cells is marked by the red bar. **b,c**, *His2AVDendra2* expression in wild-type testes. The apical tip is marked with an asterisk, the proliferating zone is outlined with white dashed lines on phase-contrast images and G2-phase gonial cells are identified with RFP. **b**, H2AVDendra2 fluorescence is apparent throughout the testis, including in the apical tip and proliferating zone, and gonial cells are circled with a dotted line. **c**, H2AVDendra2 fluorescence is intense in the nuclei of isolated and squashed gonial cells, marked with *bamGAL4*-induced RFP expression. **d,e**, *His2ADendra2* HRU expression in wild-type males. Fluorescence is absent from the proliferating zone (**d**) and from RFP-positive gonial cells (**e**) but is present in later germline cells. **f,g**, *His3Dendra2* HRU expression in wild-type males. Staining is absent throughout the germline (**f**); the few fluorescent nuclei are in somatic cells of the testis sheath (arrowhead in **f**). Squashed gonial cells are unlabeled (**g**). **h,i**, Fluorescence of the *His2ADendra2* HRU reporter in 12XWT males. This line does not carry *bamGAL4* and *UAS–RFP* constructs; the proliferating-zone and gonial cells were identified by position in the testis (dashed line in **h**) and gonial cells were identified by nuclear size and morphology

in squashes (**i**). Strong nuclear H2ADendra2 fluorescence is apparent throughout the apical tip of the testis, in gonial cells and in later stages. **j,k**, Fluorescence of the *His3Dendra2* HRU reporter in 12XWT males. This line does not carry *bamGAL4* nor *UAS–RFP*; the proliferating-zone and gonial cells were identified by position in the testis (dashed line in **j**) and gonial cells were identified by nuclear size and morphology in squashes (**k**). Strong nuclear H3Dendra2 fluorescence is apparent throughout the apical tip of the testis and in gonial cells. **l**, Fluorescence intensities of gonial nuclei for the indicated genotypes with H2AVDendra2 (*H2AV-D*; gray), H2ADendra2 (*H2A-D*; yellow) or H3Dendra2 (*H3-D*; blue) reporters. Box plots display medians and quartiles. **m**, Results of tests for derepression of *His2ADendra2* and *His3Dendra2* HRU reporters in testes with reductions in chromatin factors. *Su(var)3-9* was tested in *Su(var)3-9¹/Su(var)3-9²* homozygotes; all other factors were tested by *bamGAL4*-induced knockdown in testes. Red squares represent repression similar to the wild type, while green squares indicate derepression. **n,o**, Derepression of *His2ADendra2* (green) in whole testes (**n**) and in squashed gonial cells (**o**) with *bamGAL4*-induced RFP expression (red) and *His4* knockdown (KD). Fluorescence is absent in the apical tip of the testis but appears in the postmitotic stage where knockdown occurs. **p,q**, Derepression of *His3Dendra2* (green) in whole testes (**p**) and in squashed gonial cells (**q**) with *bamGAL4*-induced RFP expression (red) and *His4* knockdown. Fluorescence appears in the postmitotic region where knockdown occurs. Imaging of at least ten testes for each genotype–antibody combination was performed with similar results.

we examined the localization of different histones within the male germline using inducible GFP-tagged constructs[41]. Induction of a tagged H3 or tagged H3.3 histones in gonial cells broadly labeled the nuclei of these cells (Fig. 5a,b). By contrast, tagged H4 histone showed a distinct subnuclear pattern, with one major dot in each nucleus (Fig. 5c).

This dot coincided with Mxc protein at the HLB in gonial cells (Fig. 5d). As the *bamGAL4*-induced H4–GFP construct was not expressed in proliferating-zone cells, we stained testes with an antibody to histone H4. This revealed an H4 dot in proliferating-zone nuclei, including most cells actively undergoing DNA replication (Fig. 5e) and interphase

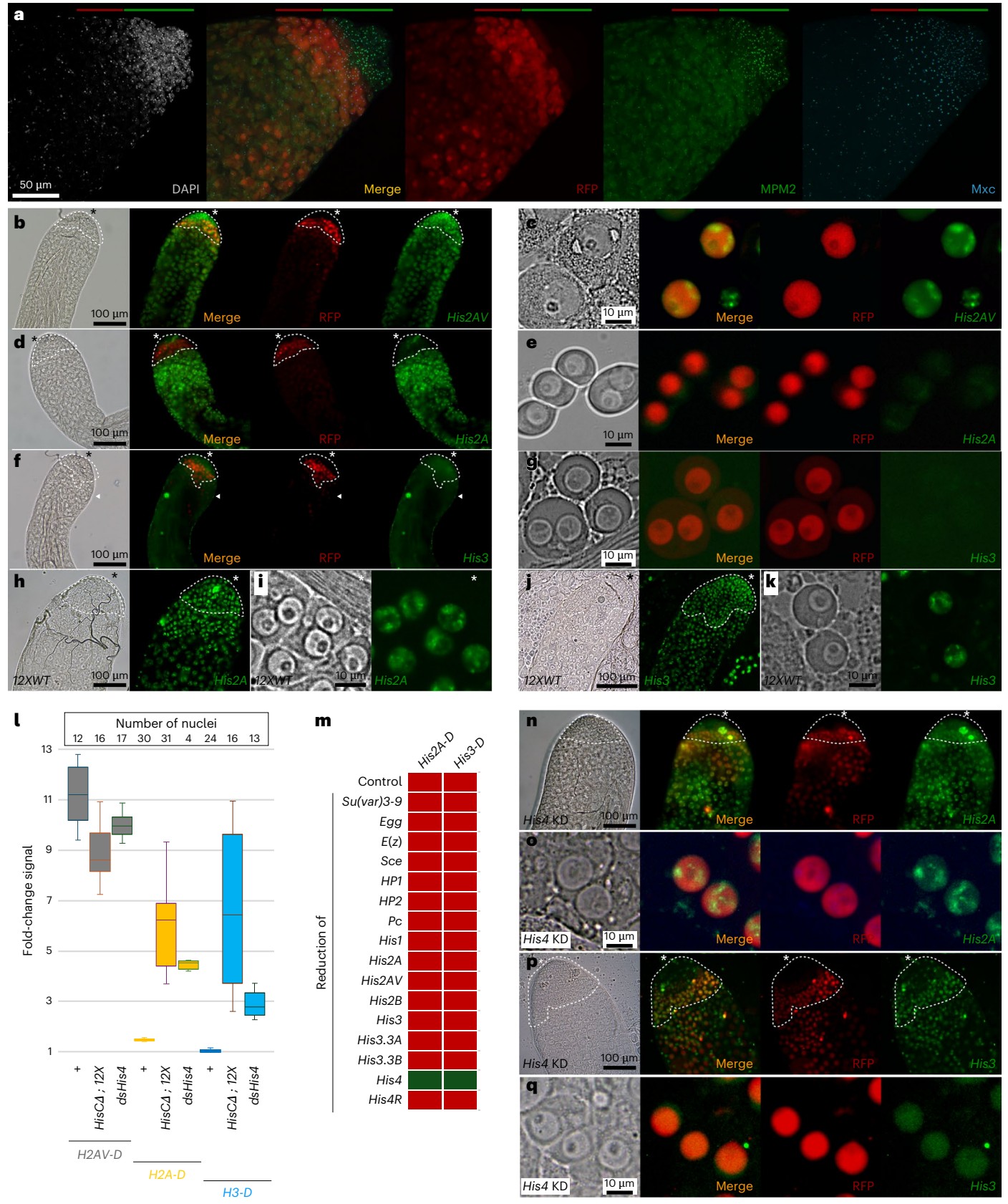

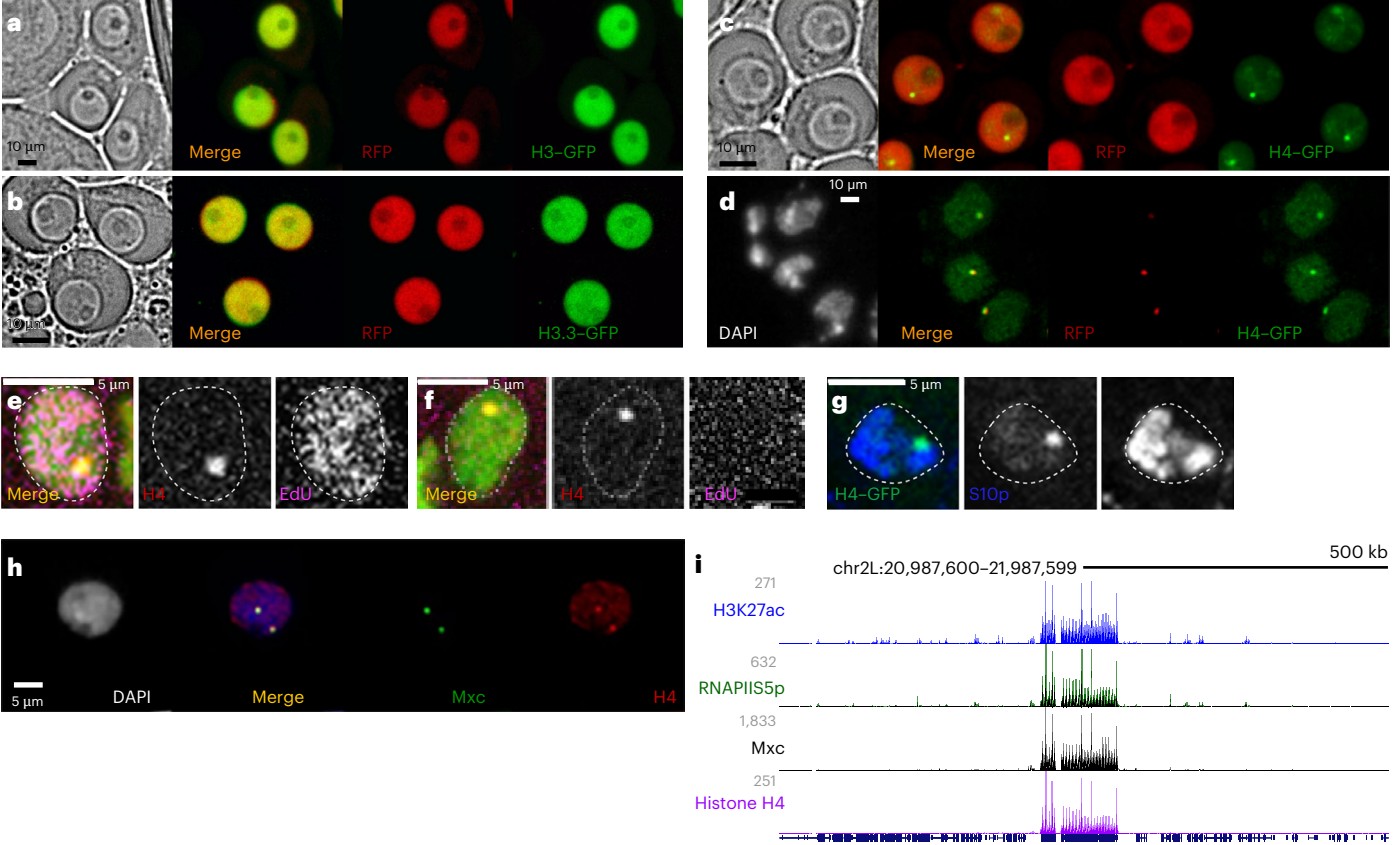

**Fig. 5 | Histone H4 is a component of the HLB in male germline cells.**
**a–c**, Fresh squashes of testes with *bamGAL4*-induced expression of GFP-tagged histones (green). Induced RFP expression (red) marks G2-phase gonial cells. Histone H3–GFP (**a**) and histone H3.3–GFP (**b**) broadly label the nuclei of gonial cells. **c**, By contrast, induced histone H4–GFP labels one bright dot with a low broad background in G2-phase nuclei. **d**, The induced histone H4–GFP (green) dot coincides with the HLB, marked by Mxc staining (red). **e,f**, Representative spermatogonial cells labeled with EdU (pink) to mark nuclei with ongoing DNA replication or in gap phase. DAPI staining is shown in green. In total, 94% (47 of 50) of S-phase cells (**e**) show focal staining of histone H4 (red) and this dot is also visible in gap-phase cells (**f**). **g**, Representative spermatogonial cell marked with the M-phase epitope H3S10p (blue). The histone H4GFP (green) dot is visible in only 7% (one of 14) of mitotic cells. Imaging of at least ten testes for each genotype–antibody combination was performed with similar results. **h**, *Kc167* nucleus stained with antibodies to histone H4 (red) and Mxc (green). Histone H4 is enriched in HLBs. Imaging of more than 100 nuclei was performed with similar results. **i**, Browser tracks of histone modifications around the histone locus in *Kc167* cells. The histone locus is enriched for the active H3K27 acetylation modification (blue), the RNAPII-S5p isoform (green), Mxc (black) and histone H4 (purple).

cells (Fig. 5f). Only 7% (one of 14) of prophase cells showed the dot (Fig. 5g), consistent with the partial disassembly of the HLB in mitosis[9]. Although histone H4 is incorporated throughout chromatin, the lack of widespread staining suggests that this antibody recognizes a part of the histone that is buried within nucleosomes and that a soluble, nonnucleosomal histone protein is in the right place to directly affect histone gene expression in both S-phase and gap-phase cells.

We examined *Drosophila* cultured *Kc167* cells to determine whether histone H4 localizes to the HLB in other cell types. In immunostained samples, histone H4 staining colocalizes with Mxc at HLBs (Fig. 5h). In chromatin profiling, histone H4 shows high signal only at the histone locus, coincident with Mxc, H3K27 acetylation and RNAPII-S5p signal (Fig. 5i). As *Kc167* cells are derived from somatic embryonic cells, this implies that nonnucleosomal histone H4 is a general component of the HLB.

### Loss of histone H4 alters HLB activity

The HLB is enriched for the initiating RNAPII-S5p isoform in embryos and in the female germline[15,16,42]. In the testis, RNAPII-S5p was distributed through nuclei and formed a bright focal spot at the HLB in proliferating-zone nuclei with bright phospho-Mxc staining and weaker focal staining in RFP-positive G2-phase gonial cells (Fig. 6a–c). This is the major isoform of RNAPII engaged at histone genes, as staining for the RNAPII-S2p isoform was broadly distributed throughout nuclei with no greater enrichment at the HLB (Fig. 6d,e). By the early primary spermatocyte stage, the HLB was no longer enriched for any RNAPII isoform. We infer that active histone loci in the proliferating zone are engaged with high levels of the RNAPII-S5p isoform and a reduced amount of this isoform persists in G2-phase gonial cells when histone genes are no longer expressed. Overall, the HLB in the testis progresses from having high levels of engaged RNAPII in the proliferating zone, to less engaged and nontranscribing RNAPII in G2-phase gonial cells and to loss of RNAPII in primary spermatocytes. Dissolution of the HLB occurs in later stages, as Mxc staining was eventually lost, as cells do not need histone gene expression as they proceed to meiosis and sperm differentiation. This progression and the spatial arrangement of the testis is an easily tractable setting to follow changes in the HLB.

The *bamGAL4*-induced H4–GFP dot was most intense in gonial cells with reduced MPM2 staining (Fig. 7a,b). In wild-type testes, the most intense spots of RNAPII-S5p staining were in the HLBs of proliferating cells with high MPM2 staining, whereas HLBs of gonial cells showed a ~60% reduction in RNAPII-S5p staining (Fig. 7c,e,g). This suggests that histone H4 has a role in limiting histone gene expression. We then examined testes where *His4* was knocked down in gonial cells. Knockdown ablated H4 staining in the HLB (Extended Data Fig. 2) and, in contrast to wild-type controls, RNAPII-S5p staining increased

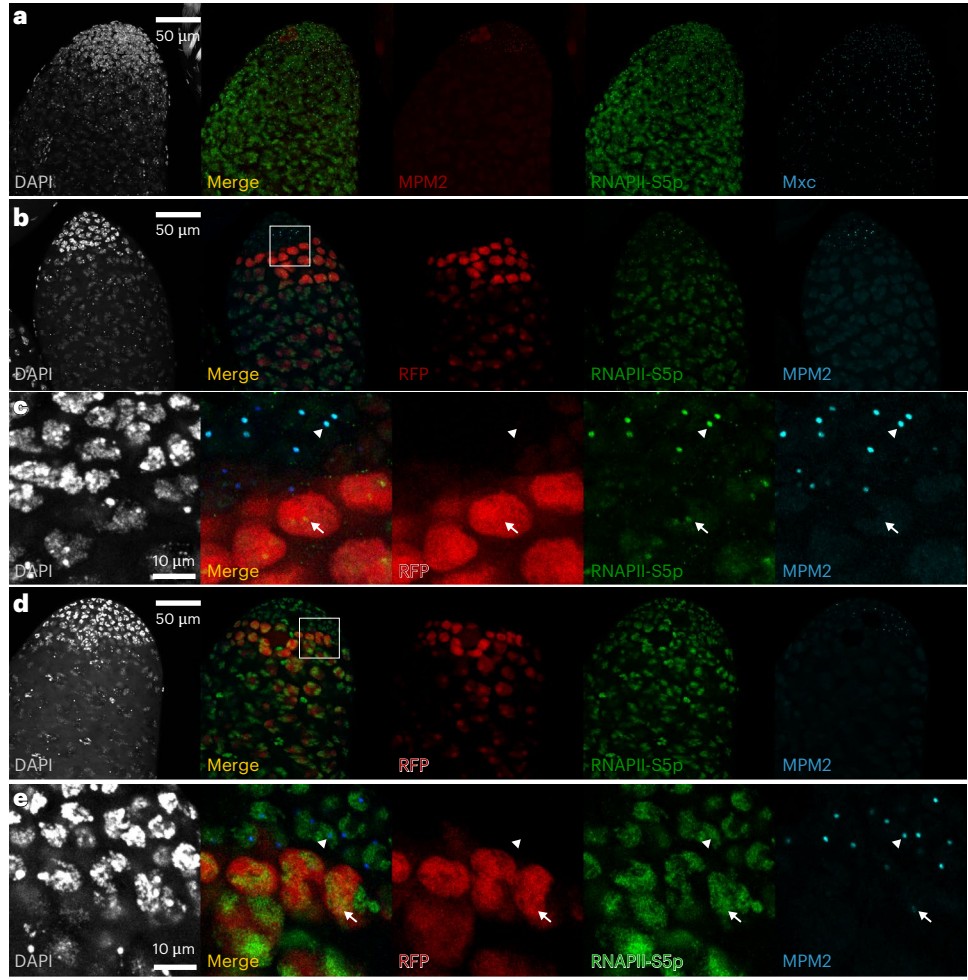

**Fig. 6 | RNAPII isoforms at the HLB in the *Drosophila* testis. a**, RNAPII-S5p (green) stains the HLB in the proliferating zone marked by phospho-Mxc staining (red) and the HLB in more distal cells. Mxc staining (blue) marks the HLB. **b**, The proliferating zone is marked with phospho-Mxc (blue) and the G2-phase gonial cells are indicated by *bamGAL4*-induced RFP (red). RNAPII-S5p staining (green) strongly stains the HLB in the proliferating zone and more weakly stains the HLB in G2-phase cells. **c**, Zoomed-in view of the boxed area in **b**. The arrowhead points to an HLB in the proliferating zone and the arrow points to an HLB in a G2-phase cell. **d**, RNAPII-S2p (green) strongly stains nuclei in the proliferating zone and in RFP-labeled G2-phase cells but is not enriched in HLBs. **e**, Zoomed-in view of the boxed area in **d**. The arrowhead points to an HLB in the proliferating zone and the arrow points to an HLB in a G2-phase cell. Imaging of at least ten testes for each genotype–antibody combination was performed with similar results.

by ~30% (Fig. 7d,f,g). These defects implicate histone H4 in the switch from active to inactive forms of the HLB.

### Histone H4 localizes to active histone gene promoters in human cells

As histone genes are repeated and we cannot distinguish between gene copies, we cannot define whether histone H4 binds all histone genes or only active or silent genes in *Drosophila*. However, histone H4 is an ancient protein and, if it has a role in limiting histone gene expression, we expect HLB localization to be conserved across species. In the human genome, the multiple copies of histone genes are not in a repeat array; instead, they are separated and scattered across four clusters, termed *HIST1–HIST4* (ref. 43), which aggregate into HLBs containing the transcription cofactor NPAT, the human homolog of Mxc[44,45]. Indeed, immunostaining of human K562 cells revealed that NPAT and histone H4 colocalized at HLBs (Fig. 8a). We then profiled the distribution of histone H4 and the active histone modification H3K27 acetylation in K562 cells and compared these to previously published profiling of NPAT[8] and RNAPII-S5p[46]. As expected, 64 canonical histone genes in the *HIST1* cluster on chromosome 6 were marked with RNAPII and with H3K27ac modifications, identifying them as active genes (Fig. 8b–d).

Each of these active histone genes was also marked with NPAT and histone H4 (Fig. 8b) and, at high resolution, these four chromatin features coincided at promoters (Fig. 8c). By contrast, the eight nontranscribed canonical histone genes lacked both NPAT and histone H4 (Fig. 8b,d), as did the five active histone variant genes that are not S-phase-regulated (Fig. 8d). Actively transcribed nonhistone genes neighboring *HIST* clusters also lacked NPAT and histone H4 signals (Fig. 8b). These results implicate histone H4 in the S-phase regulation of canonical histone gene promoters in human cells.

## Discussion

The invention of eukaryotic chromatin required the coordination of histone protein synthesis with DNA replication in S-phase cells. DNA replication and histone synthesis must be coupled because even small imbalances are detrimental, given the large amounts of chromatin duplicated during each S phase. Underproduction of histones will result in incomplete chromatin packaging and this leads to exposure and damage of new DNA[47]. Histone overproduction results in chromosome loss[48] and is cytotoxic[49]. Feedback between these two processes provides just enough histones to package newly replicated DNA[1,50]. Feedback control implies that S-phase cells measure both ongoing

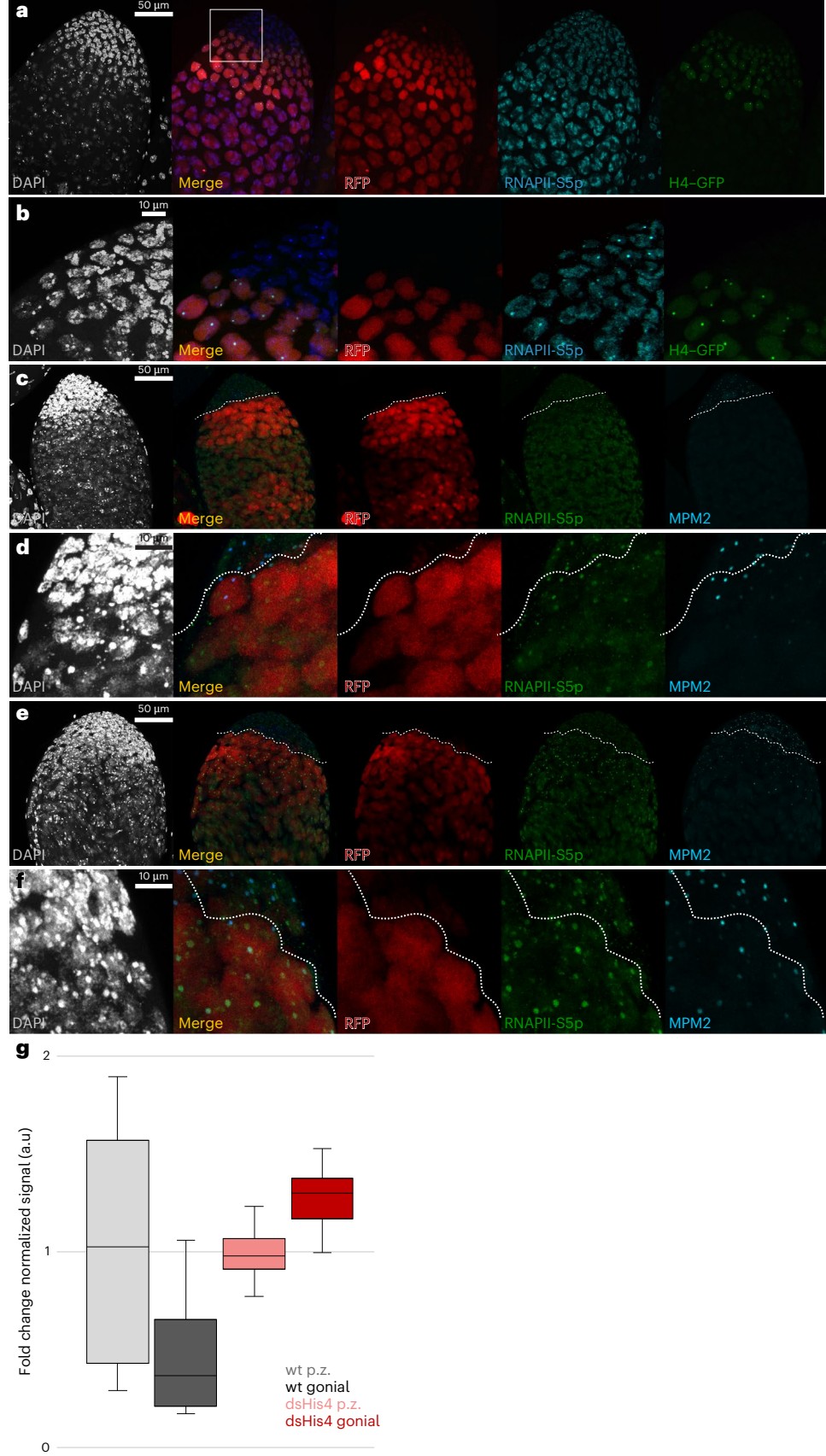

DNA replication and the need for more histones. Histone production is modulated through two main controls: CDK2-catalyzed phosphorylation of Mxc/NPAT induces transcription of canonical histone genes when cells commit to S phase[51] and cell-cycle-regulated associations between histone mRNA processing factors stabilize transcripts in S phase[52]; both of these processes take place in HLBs[53]. S-phase-induced histone mRNAs are the only protein-coding transcripts that are not polyadenylated in animals; instead, these transcripts have a terminal

**Fig. 7 | Histone H4 localizes with reduced RNAPII in HLBs. a**, Immunostaining for RNAPII-S5p (blue) in the apical tip of a testis with *bamGAL4*-induced expression of histone H4–GFP (green). G2-phase gonial cells are marked by induced RFP expression (red). The intense dot of H4–GFP coincides with the focal RNAPII-S5p signal in G2-phase gonial cells. **b**, Zoomed-in view of the box marked in **a**, showing costaining of HLBs with RNAPII-S5p and H4-GFP in RFP-positive G2-phase gonial cells. RNAPII staining in these cells is lower than that in more apical cells in the proliferating zone. **c**–**f**, Testes with *bamGAL4*-induced RFP expression (red) marking G2 gonial cells and stained for phospho-Mxc (blue) and the RNAPII-S5p isoform (green). Dashed lines demarcate the proliferating zone from G2-phase cells on the basis of RFP expression. **c**, A wild-type testis with high phospho-Mxc in the proliferating zone and low staining in G2-phase cells. The RNAPII-S5p signal in the proliferating zone is high at some HLBs and moderate at others, while it is moderate or low at HLBs in the RFP-marked cells. **d**, A zoomed-in view of the area marked in **c**, showing RNAPII-S5p signal at HLBs. **e**, A testis with

knockdown of *His4*. A high phospho-Mxc signal is apparent in the proliferating zone and occasionally persists in the RFP-labeled gonial cells. The RNAPII-S5p signal is present at HLBs of proliferating-zone cells and more intense at the HLBs in gonial cells. **f**, A zoomed-in view of the area marked in **e**, showing persistence of the RNAPII-S5p signal at HLBs in gonial cells. **g**, Quantitation of RNAPII-S5p signal in HLBs in the proliferating zone and in gonial nuclei for the indicated genotypes. p.z., proliferating zone. Box plots display medians and quartiles. RNAPII-S5p staining in wild-type testes is high in proliferating-zone cells and decreases in gonial cells. In testes with *His4* knockdown, RNAPII-S5p is high in the proliferating zone and increases in gonial cells. The intensity of RNAPII-S5p staining in knockdown gonial cells is ~2-fold that of gonial cells in the wild type. A total of 21 proliferating-zone and 33 gonial nuclei were measured in the wild-type testis, whereas 33 proliferating-zone and 34 gonial nuclei were measured in the knockdown testis. Imaging of at least ten testes for each genotype–antibody combination was performed with similar results.

stem-loop structure that is bound by SLBP, directing 3′ processing[52]. Mathematical modeling has suggested that feedback from soluble histone pools is necessary for precise coupling between DNA replication and histone synthesis[54]. Our observations suggest a simple model where soluble histone H4 protein directly represses histone gene transcription. Ongoing DNA replication and chromatin packaging use up soluble histones; however, once DNA replication ceases, soluble histones including histone H4 accumulate. Monomeric histones have been observed in cells[38] and, thus, a monomeric histone H4 might directly interact with NPAT at histone gene promoters. The majority of the NPAT and Mxc proteins are composed of intrinsically disordered domains interspersed with structured domains and both are important for self-associations and function[55,56]. We propose an interaction between the strong positive charge of the unmodified H4 N-terminal tail (N-SGRGKGGKGLGKGGAKRHRKVLR) and the strong negative charge of the unstructured regions of phosphorylated NPAT. Weak, transient charge interactions have been observed to drive chromatin binding of transcription factors[57].

The atypical 3′ ends of histone genes appear to be the ancestral organization of S-phase-induced histone genes throughout Eukaryota, because both stem-loop mRNA structures and SLBP homologs have been identified in protozoa at the base of the eukaryotic tree[58]. The evolutionary origin of this system was unclear; however, similarities to 3′ processing of transcripts in bacteria have recently been pointed out[59]. Thus, the histone 3′ processing system appears to be one of the few relics in eukaryotic genomes of their origin and may explain why these genes are sequestered in their own nuclear body. The eukaryotic histones themselves were derived from bacterial or viral proteins in the last eukaryotic common ancestor[60], with eukaryotic histone H4 being a sister lineage to both the HMfB archaeal histones and the doublet H4–H3 histones of the Nucleocytoviricota giant viruses[61], which later diversified into the four core histone subtypes[62]. Thus, it is conceivable that histone H4 has been used as a negative regulator of core histone gene expression all this time. Indeed, histone H4 is distinctive in that variants for this isotype rarely occur across eukaryotic evolution. One exception is a variant histone H4 encoded by some symbiotic bracoviruses; braconid wasps harboring this virus inject viral DNA into host moth larvae, where production of the variant histone suppresses host histone H4 mRNA production[63]. This unusual variant appears to

have weaponized the normal negative feedback loop of histone H4 on histone gene regulation for parasite life history.

Histone gene regulation has been implicated in cancer progression in humans and histone overproduction is predictive of cancer malignancy[64]. There are multiple theories to explain the initiation of a cancer, invoking genetic mutation, changes in epigenetic marks and defects in developmental signaling. Although the relative importance of these theories is now debated[65,66], in any scenario, progenitor cells must maintain a relatively undifferentiated state and proliferate. We have documented that widespread chromosome arm aneuploidies are common in cancers and the number of arm losses scales with malignancy[67]. Such aneuploidies in tumors were suggested to inhibit cellular differentiation and thereby trap anaplastic cells in a proliferating stage[68,69]. We have proposed that these events are causally linked. Histone overexpression during S phase compromises histone variant-based centromere assembly and also accelerates cell proliferation; these effects would directly cause mitotic chromosome errors and result in aneuploidies[67,69]. In support of this scenario, a recent study identified reduced cell-cycle duration as the only common feature of multiple distinct cancers, showing that tumorigenesis could be blocked by mutations affecting CDK2 activity[70]. CDK2 is the kinase that phosphorylates Mxc/NPAT for histone gene activation, linking conserved HLB regulation described here to a deeper understanding of cancer. Furthermore, a second recent study demonstrated that inhibition of PRMT5, which symmetrically dimethylates histone H4R3, results in rapid histone gene repression[71]. While the effects of anticancer PRMT5 inhibitors have been attributed to mRNA splicing defects, the very rapid effect on histone gene expression implies that this is the primary mechanism of these anticancer drugs. As such, we anticipate that detailing the mechanisms by which histone synthesis is normally restrained will provide insights into originating oncogenic events and opportunities for intervention.

## Online content

**Fig. 8 | Histone H4 localizes to the promoters of active histone genes in human K562 cells. a**, Immunostaining of human K562 cells for the HLB factor NPAT (green) and histone H4 (red). The anti-histone H4 signal localizes to each of the multiple HLBs in the nucleus. Imaging of at least 100 cells was performed with similar results. **b**, Distribution of histone H3K27 acetylation, RNAPII-Sp5, NPAT and histone H4 across a portion of the *HIST1* histone gene cluster on chromosome 6. Purple arrowheads mark three active canonical histone genes on one side of this cluster, the red arrowheads mark two inactive histone genes

and the green arrowhead marks the promoter of an adjacent active nonhistone gene. NPAT and histone H4 coincide only at active histone genes and are absent from inactive histone genes and from active nonhistone genes. **c**, Zoomed-in view showing coincidence of the H3K27 acetylation mark, RNAPII-S5p, NPAT and histone H4 at the promoters of two active histone genes. **d**, Summary of summed signal for chromatin features at 94 histone genes in the human genome. Variant histone genes are labeled in orange. Each histone isotype and variant is ordered by expression (RNAPII-S5p signal).

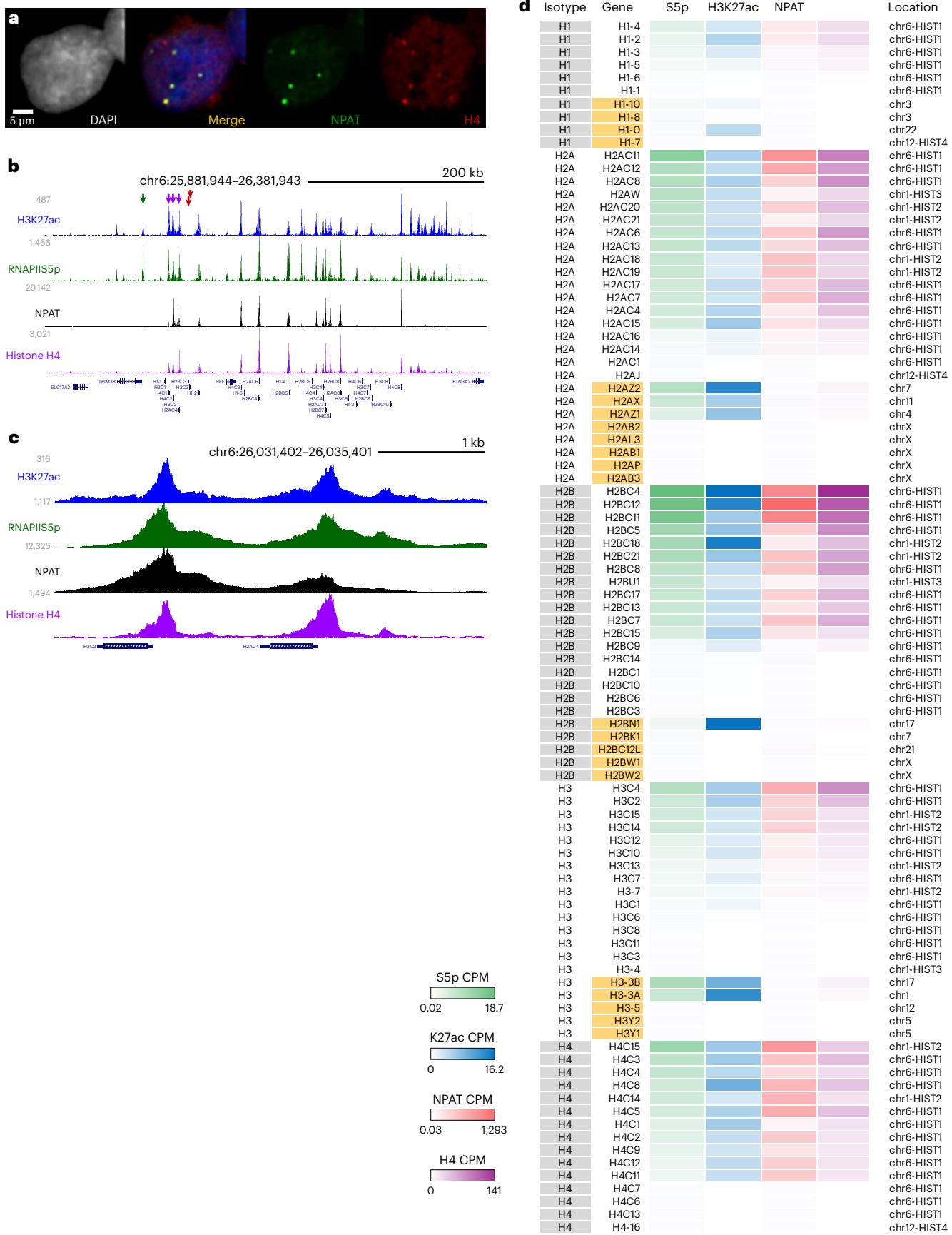

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

## Methods

### Fly strains

All crosses were performed at 25 °C. All mutations and chromosomal rearrangements used here are described in Flybase (http://www.flybase.org). The $w^{1118}$ strain was used as a wild-type control. The 12XWT strain is *w;DHis^C;12XWT* (ref. 5). The HRU reporter *His3Dendra2* was previously described[6] and the *His2ADendra2* reporter was constructed similarly. Inducible histone lines *UAS−H3−GFP* and *UAS−H3.3−GFP* were previously described[41,72] and the *UAS−H4−GFP* construct was injected into fly embryos for P element transformation[73] by BestGene. A similar UAS−H4−eGFP construct used here for some experiments was previously published[74]. Additional constructs used for cytological characterization were *y w P[bamGAL4:VP16,w^+]1/Y;P[UAS−RFP,w^+]2*. Inducible knockdown constructs and Bloomington *Drosophila* Stock Center identifiers for histones and chromatin regulators are listed in Supplementary Table 4.

### Antibodies

Antibodies used for CUT&Tag profiling and for immunocytology are listed in the Supplementary Table 4.

### Imaging fresh tissues

Dissected tissues from larvae or adults were mounted in PBS on slide and imaged by epifluorescence on an EVOS FL Auto 2 inverted microscope (Thermo Fisher Scientific) with a ×10, ×20 or ×40 objective. Details on microscope settings are provided in Supplementary Table 5. Pseudocolored images were adjusted and composited in Adobe Photoshop and Adobe Illustrator. For measuring signal intensities of imaginal discs, we used Photoshop to select the entire disc by phase-contrast imaging and summed the GFP fluorescence pixel intensity of that area in unsaturated images with identical camera settings for genotypes. For measuring signal intensities of nuclei, we used Photoshop to select gonial cell nuclei by RFP staining or by phase-contrast nuclear morphology, summed the Dendra2 fluorescence pixel intensity of that area and calculated the mean pixel intensity in unsaturated images. We normalized pixel intensities by dividing by the mean background signal in that image, such that no signal fluorescence was equal to 1. At least ten individuals of each genotype were examined.

### Imaging immunostained testes

Testes from 1-day-old adult males were dissected in PBS, incubated in Accutase (Stemcell Technologies, 07920) for 10 min at room temperature to permeabilize the tissue, fixed in 4% formaldehyde in PBS with 0.1% Triton X-100 (PBST) for 10 min, incubated twice in 0.3% sodium deoxycholate in PBST for 10 min each[75] and finally incubated with primary antibodies in A + t buffer at 4 °C overnight and then with fluorescently labeled secondary antibodies (1:200 dilution; Jackson ImmunoResearch). Testes were stained with 0.5 µg ml⁻¹ DAPI in PBS, mounted in 80% glycerol on slides and imaged on a Stellaris 8 confocal microscope (Leica) with ×20 or ×63 objectives. Details on microscope settings are provided in Supplementary Table 5. Pseudocolored maximum-intensity projections were adjusted and composited in ImageJ, Adobe Photoshop and Adobe Illustrator. For measuring signal intensities of the HLB in testes, we identified proliferating-zone nuclei from gonial cell nuclei by *bamGAL4*-induced RFP fluorescence and then used Photoshop to select the area of each HLB defined by MPM2 staining. We summed the RNAPII-S5p signal fluorescence pixel intensity of each HLB in unsaturated images and normalized scores for each testis by the mean intensity of the proliferating zone, which is not affected by knockdowns in gonial cells. At least ten testes of each genotype were imaged.

### Imaging tissue-culture cells

Drosophila *Kc167* and human K562 cells were swelled with a hypotonic 0.5% sodium citrate solution and then smashed onto glass slides in a Cytospin 4 centrifuge (Thermo). Slides were fixed with 4% formaldehyde in PBST, incubated with primary antisera in A + t buffer and then with fluorescently labeled secondary antibodies (1:200 dilution; Jackson ImmunoResearch), stained with 0.5 µg ml⁻¹ DAPI in PBS, mounted in 80% glycerol and imaged by epifluorescence on an EVOS FL Auto 2 inverted microscope (Thermo Fisher Scientific) with a ×40 objective. Pseudocolored images were adjusted and composited in Adobe Photoshop and Adobe Illustrator.

### CUT&Tag chromatin profiling

To perform CUT&Tag[8], we dissected 20 imaginal wing discs from male third-instar larvae in PBS buffer and transferred them to a tube containing Accutase (Stemcell Technologies, 07920) at 25 °C for 30 min. We then added an equal volume of 30% BSA to block proteases and ran the material through a 30G half-inch needle once to dissociate tissue. Tissue suspensions were divided across 4–8 reaction tubes and bound with BioMag Plus ConA (Bangs, 531) magnetic beads. Tissue-culture samples in media were added directly to ConA beads for binding and then lightly fixed onto beads with 0.1% formaldehyde at room temperature for 1 min. All samples were incubated with the following CUT&Tag solutions sequentially: primary antibodies diluted in Wash+ buffer (20 mM HEPES pH 7.5, 150 mM NaCl, 0.5 mM spermidine, 0.05% Triton X-100, 2 mM EDTA and 1% BSA, with Roche cOmplete protease inhibitor) overnight at 4 °C, followed by secondary antibodies (in Wash+ buffer) for 1 h at room temperature and then pAGTn5 (Epicypher 15-1017) in 300Wash+ buffer (20 mM HEPES pH 7.5, 300 mM NaCl, 0.5 mM spermidine and 0.05% Triton X-100 with cOmplete protease inhibitor) for 1 h. After one wash with 300Wash+ buffer, samples were incubated in 300Wash+ buffer supplemented with 10 mM MgCl₂ for 1 h at 25 °C to tagment chromatin. Tissue-culture samples were tagmented in CUTAC buffer (10 mM TAPS pH 8.5 and 20% DMF) supplemented with 5 mM MgCl₂ to enhance tagmentation efficiency. Samples were washed with 10 mM TAPS pH 8.5 and DNA was released with 0.1% SDS, 0.012 U per µl thermolabile protease K (New England Biolabs, P8111S) in 10 mM TAPS at 37 °C and inactivated at 55 °C. Libraries were enriched with 14 cycles of PCR according to a previous protocol[8] and sequenced in dual-indexed paired-end 50-bp mode on the Illumina NextSeq 2000 or NovaSeq platforms at the Fred Hutchinson Cancer Center Genomics Shared Resource. Paired-end reads were mapped to this assembly using Bowtie2 using parameters such as '--end-to-end --very-sensitive --no-mixed --no-discordant -q --phred33 -I 10 -X 700'.

### Gene score tables

To summarize the enrichment of profiling features across histones, we counted mapped reads from the start to the end of each gene in BAM files using subreads/feature_counts with option ' _o'. Counts for each replication-coupled histone were summed and counts for all genes were scaled by total mapped reads to give counts per million reads. These values are provided in Supplementary Table 2.

### Genomic display

Files of mapped reads were converted to genome coverage with bedtools/bamcoverage[76] and displayed in the UCSC genome browser[77]. Selected regions were exported in PDF format and formatted with Adobe Illustrator.

### Reporting summary

Further information on research design is available in the Nature Portfolio Reporting Summary linked to this article.

## Data availability

Sequencing data were deposited to the Gene Expression Omnibus (GEO) under accession code GSE280833. Data for NPAT profiling in K562 cells (SH_Hs_NPA1_20190217, SH_Hs_NPA2_20190217, SH_Hs_NPA4_20190217, SH_Hs_NPA8_20190217, SH_Hs_NPB1_20190217,

SH_Hs_NPB2_20190217, SH_Hs_NPB4_20190217 and SH_Hs_NPB8_20190217) were previously published[8] and were merged here into one file (SH_Hs_NPB4_20190217). Data for RNAPII-S5p profiling in K562 cells (SH_Hs_K5xlin_PolS5P_3cy_0320, SH_Hs_K5xlin_PolS5P_6cy, SH_Hs_K5xlin_PolS5P_9cy_0320, SH_Hs_K5xlin_PolS5P_12cy_0320 and SH_Hs_K5xlin_PolS5P_20k_0320) were previously published[46] and were merged here in one file (SH_Hs_K5xlin_PolS5P_20k_0320). Source data are provided with this paper.

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

## Acknowledgements

We thank A. Amodeo and Y. Shindo for histone–Dendra2 reporter lines including two unpublished lines (*His2ADendra2* HRU reporter and *His2AVDendra2* reporter) and B. Duronio for histone locus rescue lines. We thank C. Codomo and J. Henikoff for technical and bioinformatic support and P. Talbert for pointing out the importance of viral histone H4 protein. Work was supported by funding from the Howard Hughes Medical Institute awarded to S.H. The funders had no role in study design, data collection and analysis, decision to publish or preparation of the manuscript.

## Author contributions

K.A. and S.H. conceptualized the study. K.A., M.W., B.N.T. and V.V. performed the experiments. K.A., S.H., and V.V. performed the data analysis. K.A. wrote the manuscript. K.A., S.H. and X.C. reviewed and edited the manuscript. All authors approved the manuscript.

## Competing interests

S.H. and K.A. have filed patent applications on related work. The remaining authors declare no competing interests.

## Additional information

**Extended data** is available for this paper at https://doi.org/10.1038/s41594-025-01731-1.

**Correspondence and requests for materials** should be addressed to Kami Ahmad or Steven Henikoff.

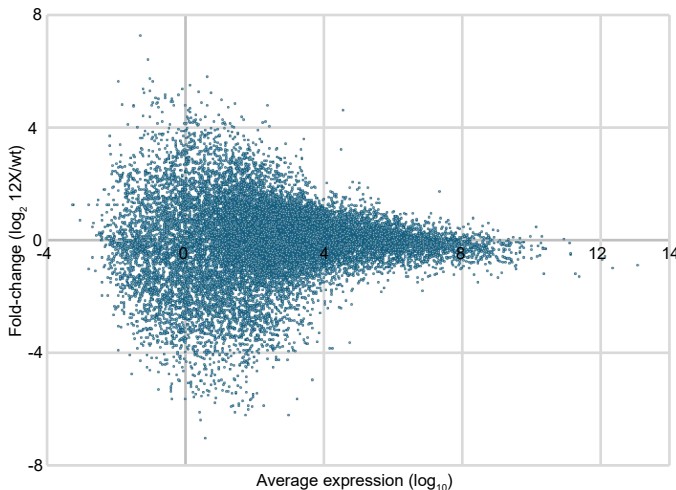

**Extended Data Fig. 1 | MA plot of RNAPII-S2p signal at genes in wildtype and *HisCΔ ; 12XWT* wing imaginal discs.** Values are presented in Supplementary Table 3.

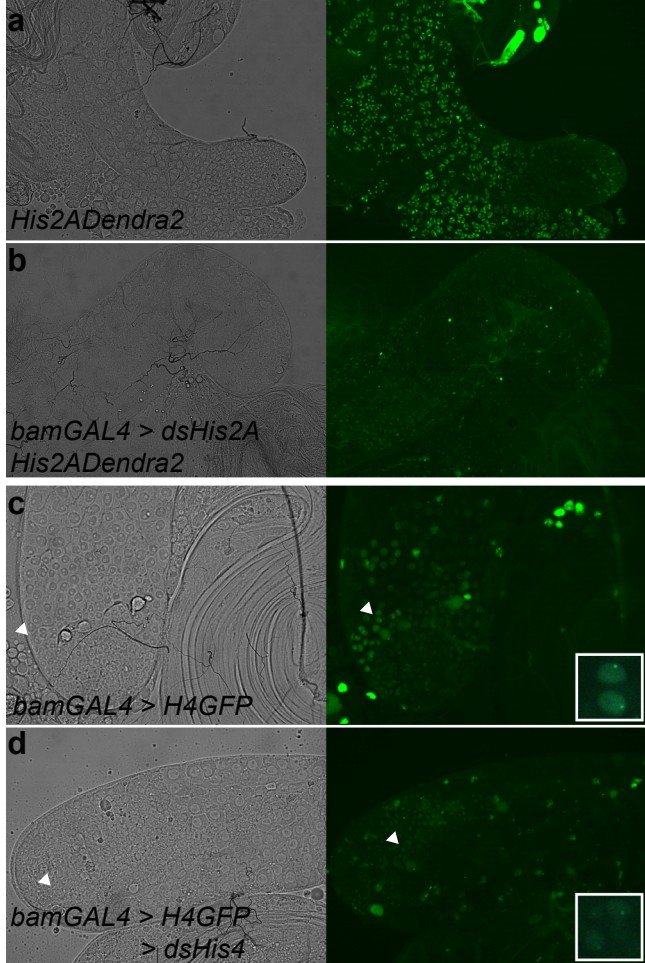

100 μm

**Extended Data Fig. 2 | Knockdown efficiency of two histone RNAi constructs in the testis. a**, *His2ADendra2* is expressed in the wild-type testis. **b**, *bamGAL4*-induced knockdown of *His2A* reduces *His2ADendra2* expression. **c**, *bamGAL4*-induced H4-GFP labels nuclei in the testis. The arrowhead indicates H4-GFP fluorescence at the HLB, shown in the inset zoom. **d**, *bamGAL4*-induced H4-GFP labeling is reduced by induced *His4* knockdown. Residual staining at an HLB is shown in the inset. Imaging was performed on 5 testes of each genotype with similar results.

# Reporting Summary

## Statistics

For all statistical analyses, confirm that the following items are present in the figure legend, table legend, main text, or Methods section.

| n/a | Confirmed | |
|---|---|---|
| ☐ | ☒ | The exact sample size (*n*) for each experimental group/condition, given as a discrete number and unit of measurement |
| ☒ | ☐ | A statement on whether measurements were taken from distinct samples or whether the same sample was measured repeatedly |
| ☒ | ☐ | The statistical test(s) used AND whether they are one- or two-sided<br>*Only common tests should be described solely by name; describe more complex techniques in the Methods section.* |
| ☒ | ☐ | A description of all covariates tested |
| ☒ | ☐ | A description of any assumptions or corrections, such as tests of normality and adjustment for multiple comparisons |
| ☒ | ☐ | A full description of the statistical parameters including central tendency (e.g. means) or other basic estimates (e.g. regression coefficient) AND variation (e.g. standard deviation) or associated estimates of uncertainty (e.g. confidence intervals) |
| ☒ | ☐ | For null hypothesis testing, the test statistic (e.g. *F*, *t*, *r*) with confidence intervals, effect sizes, degrees of freedom and *P* value noted<br>*Give P values as exact values whenever suitable.* |
| ☒ | ☐ | For Bayesian analysis, information on the choice of priors and Markov chain Monte Carlo settings |
| ☒ | ☐ | For hierarchical and complex designs, identification of the appropriate level for tests and full reporting of outcomes |
| ☒ | ☐ | Estimates of effect sizes (e.g. Cohen's *d*, Pearson's *r*), indicating how they were calculated |

*Our web collection on statistics for biologists contains articles on many of the points above.*

## Software and code

Policy information about availability of computer code

| Data collection | The size distributions and molar concentration of libraries were determined using an Agilent 4200 TapeStation. Barcoded CUT&RUN libraries were pooled at approximately equimolar concentration for sequencing. Paired-end 50×50 bp sequencing on the Illumina NextSeq 2000 platform was performed by the Fred Hutchinson Cancer Center Genomics Shared Resources. This yielded 1-40 million reads per antibody. To remove adapter sequences, we preprocessed the reads using cutadapt version 2.9 with parameters -j 8 --nextseq-trim 20 -m 20 -a AGATCGGAAGAGCACACGTCTGAACTCCAGTCA -A AGATCGGAAGAGCGTCGTGTAGGGAAAGAGTGT -Z. Paired-end reads were then aligned to the UCSC hg38 human genome build using Bowtie2 version 2.4.4 with parameters --very-sensitive-local --soft-clipped-unmapped-tlen --dovetail --no-mixed --no-discordant -q --phred33 -I 10 -X 1000. |
|---|---|
| Data analysis | Bedtools, featureCounts |

For manuscripts utilizing custom algorithms or software that are central to the research but not yet described in published literature, software must be made available to editors and reviewers. We strongly encourage code deposition in a community repository (e.g. GitHub). See the Nature Portfolio guidelines for submitting code & software for further information.

## Data

Policy information about availability of data

All manuscripts must include a data availability statement. This statement should provide the following information, where applicable:

- Accession codes, unique identifiers, or web links for publicly available datasets
- A description of any restrictions on data availability
- For clinical datasets or third party data, please ensure that the statement adheres to our policy

All sequencing data have been deposited as bigwig files in Gene Expression Omnibus under the accession code GSE280833.

## Research involving human participants, their data, or biological material

Policy information about studies with human participants or human data. See also policy information about sex, gender (identity/presentation), and sexual orientation and race, ethnicity and racism.

| | |
|---|---|
| Reporting on sex and gender | n/a |
| Reporting on race, ethnicity, or other socially relevant groupings | n/a |
| Population characteristics | n/a |
| Recruitment | n/a |
| Ethics oversight | n/a |

Note that full information on the approval of the study protocol must also be provided in the manuscript.

# Field-specific reporting

Please select the one below that is the best fit for your research. If you are not sure, read the appropriate sections before making your selection.

☒ Life sciences    ☐ Behavioural & social sciences    ☐ Ecological, evolutionary & environmental sciences

For a reference copy of the document with all sections, see nature.com/documents/nr-reporting-summary-flat.pdf

# Life sciences study design

All studies must disclose on these points even when the disclosure is negative.

| | |
|---|---|
| Sample size | At least 10 examples of each genotype was characterized for each immunostaining experiment. |
| Data exclusions | No data were excluded |
| Replication | Experiments were repeated at least 3 times for chromatin profiling, and at least 5 times for immunostaining experiments. |
| Randomization | n/a. Multiple individuals of each genotype were characterized. |
| Blinding | n/a. Experiments characterized genotypes. |

# Reporting for specific materials, systems and methods

We require information from authors about some types of materials, experimental systems and methods used in many studies. Here, indicate whether each material, system or method listed is relevant to your study. If you are not sure if a list item applies to your research, read the appropriate section before selecting a response.

## Materials & experimental systems

| n/a | Involved in the study |
|---|---|
| ☐ | ☒ Antibodies |
| ☐ | ☒ Eukaryotic cell lines |
| ☐ | ☐ Palaeontology and archaeology |
| ☐ | ☒ Animals and other organisms |
| ☐ | ☐ Clinical data |
| ☐ | ☐ Dual use research of concern |
| ☐ | ☐ Plants |

## Methods

| n/a | Involved in the study |
|---|---|
| ☐ | ☒ ChIP-seq |
| ☐ | ☐ Flow cytometry |
| ☐ | ☐ MRI-based neuroimaging |

## Antibodies

| | |
|---|---|
| Antibodies used | anti-RNAPII (mouse) Covance Research Products MMS-126R<br>anti-RNAPII-S5p (rabbit) Cell Signalling Technology D9N5I<br>anti-RNAPII-S2p (rabbit) Cell Signalling Technology E1Z3G<br>anti-Mxc (guinea pig) RJ Duronio Mxc-C-1<br>anti-Mute (guinea pig) RJ Duronio Mute<br>anti-H3K9me1 (rabbit) Epicypher 13-0029<br>anti-H3K9me2 (mouse) EMD Millipore 05-1249<br>anti-H3K9me3 (rabbit) Abcam ab8898<br>anti-H3K27me3 (rabbit) Cell Signalling Technology C36B11<br>anti-uH2A (rabbit) Cell Signalling Technology D27C4<br>anti-H3K4me1 (rabbit) Epicypher 13-0057<br>anti-H3K4me2 (rabbit) Epicypher 13-0027<br>anti-H3K4me3 (rabbit) Epicypher 13-0060<br>anti-H3K27ac (rabbit) Epicypher 13-0059<br>anti-H3K36me3 (rabbit) Epicypher 13-0058<br>anti-MPM2 (mouse) DAKO M3514<br>anti-GFP (rabbit) Cell Signalling Technology D5.1<br>anti-GFP (mouse) Thermo Fisher Scientific 3E6<br>anti-rabbit IgG (guinea pig) Antibodies Online ABIN101961<br>anti-mouse IgG (rabbit) Abcam ab46540<br>anti-guinea pig IgG (rabbit) Thermo Fisher Scientific PA1-28549<br>anti-rabbit-FITC (goat) Jackson ImmunoResearch 111-095-144<br>anti-mouse-FITC (goat) Jackson ImmunoResearch 115-095-166<br>anti-mouse-RRX (goat) Jackson ImmunoResearch 115-295-166<br>anti-mouse-Cy5 (goat) Jackson ImmunoResearch 115-175-166<br>anti-guinea pig-TRITC (goat) Jackson ImmunoResearch 706-025-148<br>anti-guinea pig-Cy5 (donkey) Jackson ImmunoResearch 706-175-148<br>anti-histone H4 (mouse) Abcam ab31830<br>anti-histone H3-S10p (mouse) Abcam Ab14955<br>anti-NPAT (rabbit) Thermo Fisher Scientific PA565419 |
| Validation | All commercial antibodies were validated by the suppliers. Anti-Mute and anti-Mxc antibodies were validated by R. Duronio (UNC) |

## Eukaryotic cell lines

Policy information about cell lines and Sex and Gender in Research

| | |
|---|---|
| Cell line source(s) | Kc167, Drosophila melanogaster female. K562, human female |
| Authentication | K562 cells were authenticated by ATCC by STR analysis. |
| Mycoplasma contamination | all cell lines tested negative for mycoplasma. |
| Commonly misidentified lines<br>(See ICLAC register) | n/a |

## Palaeontology and Archaeology

| | |
|---|---|
| Specimen provenance | *Provide provenance information for specimens and describe permits that were obtained for the work (including the name of the issuing authority, the date of issue, and any identifying information). Permits should encompass collection and, where applicable, export.* |
| Specimen deposition | *Indicate where the specimens have been deposited to permit free access by other researchers.* |

| Dating methods | *If new dates are provided, describe how they were obtained (e.g. collection, storage, sample pretreatment and measurement), where they were obtained (i.e. lab name), the calibration program and the protocol for quality assurance OR state that no new dates are provided.* |

☐ Tick this box to confirm that the raw and calibrated dates are available in the paper or in Supplementary Information.

| Ethics oversight | *Identify the organization(s) that approved or provided guidance on the study protocol, OR state that no ethical approval or guidance was required and explain why not.* |

Note that full information on the approval of the study protocol must also be provided in the manuscript.

# Animals and other research organisms

Policy information about studies involving animals; ARRIVE guidelines recommended for reporting animal research, and Sex and Gender in Research

| Laboratory animals | Drosophila melanogaster |
| Wild animals | The study did not use wild animals. |
| Reporting on sex | Findings were observed in both male and female cells. |
| Field-collected samples | The study did not involve animals collected in the field. |
| Ethics oversight | No oversight was needed for work with Drosophila melanogaster or with the cell lines used. |

Note that full information on the approval of the study protocol must also be provided in the manuscript.

# Clinical data

Policy information about clinical studies

All manuscripts should comply with the ICMJE guidelines for publication of clinical research and a completed CONSORT checklist must be included with all submissions.

| Clinical trial registration | *Provide the trial registration number from ClinicalTrials.gov or an equivalent agency.* |
| Study protocol | *Note where the full trial protocol can be accessed OR if not available, explain why.* |
| Data collection | *Describe the settings and locales of data collection, noting the time periods of recruitment and data collection.* |
| Outcomes | *Describe how you pre-defined primary and secondary outcome measures and how you assessed these measures.* |

# Dual use research of concern

Policy information about dual use research of concern

## Hazards

Could the accidental, deliberate or reckless misuse of agents or technologies generated in the work, or the application of information presented in the manuscript, pose a threat to:

No | Yes

☐ | ☐ Public health

☐ | ☐ National security

☐ | ☐ Crops and/or livestock

☐ | ☐ Ecosystems

☐ | ☐ Any other significant area

## Experiments of concern

Does the work involve any of these experiments of concern:

| No | Yes | |
|----|-----|---|
| ☐ | ☐ | Demonstrate how to render a vaccine ineffective |
| ☐ | ☐ | Confer resistance to therapeutically useful antibiotics or antiviral agents |
| ☐ | ☐ | Enhance the virulence of a pathogen or render a nonpathogen virulent |
| ☐ | ☐ | Increase transmissibility of a pathogen |
| ☐ | ☐ | Alter the host range of a pathogen |
| ☐ | ☐ | Enable evasion of diagnostic/detection modalities |
| ☐ | ☐ | Enable the weaponization of a biological agent or toxin |
| ☐ | ☐ | Any other potentially harmful combination of experiments and agents |

## Plants

| | |
|---|---|
| Seed stocks | *Report on the source of all seed stocks or other plant material used. If applicable, state the seed stock centre and catalogue number. If plant specimens were collected from the field, describe the collection location, date and sampling procedures.* |
| Novel plant genotypes | *Describe the methods by which all novel plant genotypes were produced. This includes those generated by transgenic approaches, gene editing, chemical/radiation-based mutagenesis and hybridization. For transgenic lines, describe the transformation method, the number of independent lines analyzed and the generation upon which experiments were performed. For gene-edited lines, describe the editor used, the endogenous sequence targeted for editing, the targeting guide RNA sequence (if applicable) and how the editor was applied.* |
| Authentication | *Describe any authentication procedures for each seed stock used or novel genotype generated. Describe any experiments used to assess the effect of a mutation and, where applicable, how potential secondary effects (e.g. second site T-DNA insertions, mosiacism, off-target gene editing) were examined.* |

## ChIP-seq

### Data deposition

☒ Confirm that both raw and final processed data have been deposited in a public database such as GEO.

☐ Confirm that you have deposited or provided access to graph files (e.g. BED files) for the called peaks.

| | |
|---|---|
| Data access links<br>*May remain private before publication.* | Sequencing data is available for review under the GEO accession GSE280833 at www.ncbi.nlm.nih.gov/geo/query/acc.cgi?acc=GSE280833, enter token wnafcsoilhwxpuz in the box. |
| Files in database submission | GSM8606309 w_RNAPII_(220119_BT_Dm_BT542_P222)<br>GSM8606310 w_H3K36ne3_R1_(220406_BT_Hs_BT826_H4)<br>GSM8606311 w_uH2A_R1_(220406_BT_Hs_BT827_H106)<br>GSM8606312 w_H3K27me3_R1_(220819_BT_Dm_BT1173_H128)<br>GSM8606313 w_H3K9me3_R1_(220819_BT_Dm_BT1175_H28)<br>GSM8606314 w_H3K9me1_R1_(230404_BT_Dm_BT1777_H126)<br>GSM8606315 w_H3K9me2_R1_(230404_BT_Dm_BT1778_H1)<br>GSM8606316 w_RNAPII-S5p_R1_(230404_BT_Dm_BT1781_P51)<br>GSM8606317 12XWT_RNAPII-S5p_R1_(230522_BT_Dm_BT1933_P51)<br>GSM8606318 12XWT_H3K9me2_R1_(230522_BT_Dm_BT1936_H1)<br>GSM8606319 12XWT_H3K27me3_R1_(230522_BT_Dm_BT1939_H128)<br>GSM8606320 12XWT_H3K27ac_R1_(230522_BT_Dm_BT1940_217)<br>GSM8606321 12XWT_H3K36me3_R1_(230522_BT_Dm_BT1942_283)<br>GSM8606322 12XWT_uH2A_R1_(230522_BT_Dm_BT1943_H106)<br>GSM8606323 12XWT_Mute_R1_(230522_BT_Dm_BT1945_mute)<br>GSM8606324 12XWT_H3K9me2_R2_(230628_BT_Dm_BT1965_H1)<br>GSM8606325 12XWT_RNAPII-S5p_R2_(230704_BT_Dm_BT1993_P51)<br>GSM8606326 12XWT_H3K9me2_R3_(230704_BT_Dm_BT1994_H1)<br>GSM8606327 12XWT_uH2A_R2_(230704_BT_Dm_BT1996_H106)<br>GSM8606328 12XWT_H3K27ac_R2_(230704_BT_Dm_BT1997_217)<br>GSM8606329 12XWT_Mxc_R1_(230704_BT_Dm_BT1998_268)<br>GSM8606330 12XWT_Mxc_R2_(230704_BT_Dm_BT1999_269)<br>GSM8606331 12XWT_Mute_R2_(230704_BT_Dm_BT2000_mute)<br>GSM8606332 w_Mute_R1_(240222_BT_Dm_2579_mute)<br>GSM8606333 12XWT_Mute_R3_(240525_BT_Dm_3052)<br>GSM8606334 12XWT_Mxc_R3_(240525_BT_Dm_3053)<br>GSM8606335 12XWT_RNAPII-S2p_R1_(240525_BT_Dm_3055)<br>GSM8606336 12XWT_RNAPII-S2p_R2_(240525_BT_Dm_3056)<br>GSM8606337 12XWT_H3K9me3_R1_(240525_BT_Dm_3057) |

GSM8606338 12XWT_H3K9me3_R2_(240525_BT_Dm_3058)
GSM8606339 12XWT_H3K4me1_R1_(240525_BT_Dm_3059)
GSM8606340 12XWT_H3K4me2_R1_(240525_BT_Dm_3060)
GSM8606341 12XWT_H3K4me3_R1_(240525_BT_Dm_3061)
GSM8606342 12XWT_H3K4me3_R2_(240525_BT_Dm_3062)
GSM8606343 12XWT_H3K27ac_R3_(240525_BT_Dm_3063)
GSM8606344 w_Mxc_R1_(240525_BT_Dm_3065)
GSM8606345 w_RNAPII-S2p_R1_(240525_BT_Dm_3067)
GSM8606346 w_H3K4me2_R1_(240525_BT_Dm_3068)
GSM8606347 w_H3K27ac_R1_(240525_BT_Dm_3069)
GSM8606348 w_H3K4me1_R1_(240525_BT_Dm_3072)
GSM8606349 w_H3K4me3_R1_(240525_BT_Dm_3074)
GSM8988491 Kc_Mxc_(240709_BT_Dm_3195)
GSM8988492 Kc_RNAPIIS5p_(240820_BT_Dm_3417)
GSM8988493 Kc_H3K27ac_(241125_BT_Dm_3505)
GSM8988494 Kc_histoneH4_(241125_BT_Dm_3510)
GSM8988495 K562_H3K27ac_R1_(241125_BT_Hs_3512)
GSM8988496 K562_H3K27ac_R2_(241125_BT_Hs_3513)
GSM8988497 K562_H3K27ac_R3_(241125_BT_Hs_3514)
GSM8988498 K562_histoneH4_R1_(241125_BT_Hs_3516)
GSM8988499 K562_histoneH4_R2_(241125_BT_Hs_3517)
GSM8988500 K562_histoneH4_R3_(241125_BT_Hs_3518)

**Genome browser session** (e.g. UCSC)

n/a

# Methodology

**Replicates**

A least two replicates were performed for each profile.

**Sequencing depth**

All sequencing was PE50:
Sample ID species genotype antibody epitope  mapped reads
BT542 D.melanogaster w P222 Rpb1  19,594,989
BT826 D.melanogaster w H4 H3K36me3  8,952,249
BT827 D.melanogaster w H106 uH2A  8,067,594
BT1173 D.melanogaster w H128 H3K27me3  5,731,799
BT1175 D.melanogaster w H28 H3K9me3  5,201,319
BT1777 D.melanogaster w H126 H3K9me1  3,095,758
BT1778 D.melanogaster w H1 H3K9me2  1,824,927
BT1781 D.melanogaster w P51 RNAPIIS5p  3,055,368
BT1933 D.melanogaster 12XWT P51 RNAPIIS5p  2,349,892
BT1936 D.melanogaster 12XWT H1 H3K9me2  11,809,042
BT1939 D.melanogaster 12XWT H128 H3K27me3  13,191,375
BT1940 D.melanogaster 12XWT 217 H3K27ac  13,722,592
BT1942 D.melanogaster 12XWT 283 H3K36me3  12,842,116
BT1943 D.melanogaster 12XWT H106 uH2A  12,108,154
BT1945 D.melanogaster 12XWT MUTE Mute  760,306
BT1965 D.melanogaster 12XWT H1 H3K9me2  1,785,978
BT1993 D.melanogaster 12XWT P51 RNAPIIS5p  1,949,469
BT1994 D.melanogaster 12XWT H1 H3K9me2  6,829,907
BT1996 D.melanogaster 12XWT H106 uH2A  7,297,519
BT1997 D.melanogaster 12XWT 217 H3K27ac  8,626,044
BT1998 D.melanogaster 12XWT 268 Mxc  1,384,718
BT1999 D.melanogaster 12XWT 269 Mxc  2,751,886
BT2000 D.melanogaster 12XWT Mute Mute  1,532,936
BT2579 D.melanogaster w Mute Mute  4,918,967
BT3052 D.melanogaster 12XWT Mute Mute  3,360,968
BT3053 D.melanogaster 12XWT 268 Mxc  417,996
BT3055 D.melanogaster 12XWT P72 RNAPIIS2p  1,541,295
BT3056 D.melanogaster 12XWT P72 RNAPIIS2p  3,497,752
BT3057 D.melanogaster 12XWT H28 H3K9me3  4,237,975
BT3058 D.melanogaster 12XWT H28 H3K9me3  2,825,340
BT3059 D.melanogaster 12XWT 281 H3K4me1  4,578,654
BT3060 D.melanogaster 12XWT H14 H3K4me2  4,910,423
BT3061 D.melanogaster 12XWT 289 H3K4me3  2,429,885
BT3062 D.melanogaster 12XWT 289 H3K4me3  5,472,314
BT3063 D.melanogaster 12XWT 290 H3K27ac  6,295,838
BT3065 D.melanogaster w 269 Mxc  991,254
BT3067 D.melanogaster w P72 RNAPIIS2p  3,187,095
BT3068 D.melanogaster w H14 H3K4me2  7,731,962
BT3069 D.melanogaster w 217 H3K27ac  8,449,152
BT3072 D.melanogaster w 281 H3K4me1  5,498,442
BT3074 D.melanogaster w 289 H3K4me3  6,092,071
BT3505 D.melanogaster Kc167 cells 290 H3K27ac  8,935,323
BT3417 D.melanogaster Kc167 cells P51 RNAPIIS5p  4,553,069
BT3195 D.melanogaster Kc167 cells 269 Mxc  7,576,135

BT3510 D.melanogaster Kc167 cells H157 histone H4   6,152,565
BT3512 H. sapiens K562 cells 290 H3K27ac   7,015,990
BT3513 H. sapiens K562 cells 290 H3K27ac   6,030,022
BT3514 H. sapiens K562 cells 290 H3K27ac   9,469,857
BT3516 H. sapiens K562 cells H157 histone H4   5,799,455
BT3517 H. sapiens K562 cells H157 histone H4   6,694,997
BT3518 H. sapiens K562 cells H157 histone H4   5,202,544
SH_Hs_K5xlin_PolS5P_1k_0320.DTmarked H. sapiens K562 cells P51 RNAPIIS5p   7,874,209
SH_Hs_K5xlin_PolS5P_2k_0320.DTmarked H. sapiens K562 cells P51 RNAPIIS5p   6,663,749
SH_Hs_K5xlin_PolS5P_3cy_0320.DTmarked H. sapiens K562 cells P51 RNAPIIS5p   4,355,749
SH_Hs_K5xlin_PolS5P_5k_0320.DTmarked H. sapiens K562 cells P51 RNAPIIS5p   6,925,587
SH_Hs_K5xlin_PolS5P_6cy_0320.DTmarked H. sapiens K562 cells P51 RNAPIIS5p   7,329,410
SH_Hs_K5xlin_PolS5P_9cy_0320.DTmarked H. sapiens K562 cells P51 RNAPIIS5p   10,037,905
SH_Hs_K5xlin_PolS5P_10k_0320.DTmarked H. sapiens K562 cells P51 RNAPIIS5p   8,990,835
SH_Hs_K5xlin_PolS5P_12cy_0320.DTmarked H. sapiens K562 cells P51 RNAPIIS5p   8,376,833
SH_Hs_K5xlin_PolS5P_20k_0320.DTmarked H. sapiens K562 cells P51 RNAPIIS5p   11,086,923
SH_Hs_K5xlin_PolS5P_100_0320.DTmarked H. sapiens K562 cells P51 RNAPIIS5p   3,714,156
SH_Hs_K5xlin_PolS5P_200_0320.DTmarked H. sapiens K562 cells P51 RNAPIIS5p   3,937,377
SH_Hs_K5xlin_PolS5P_500_0320.DTmarked H. sapiens K562 cells P51 RNAPIIS5p   6,632,722
SH_Hs_NPA1_20190217.DTmarked H. sapiens K562 cells P281 NPAT   2,429,157
SH_Hs_NPA2_20190217.DTmarked H. sapiens K562 cells P281 NPAT   2,002,911
SH_Hs_NPA4_20190217.DTmarked H. sapiens K562 cells P281 NPAT   1,840,725
SH_Hs_NPA8_20190217.DTmarked H. sapiens K562 cells P281 NPAT   920,075
SH_Hs_NPB1_20190217.DTmarked H. sapiens K562 cells P281 NPAT   2,648,637
SH_Hs_NPB2_20190217.DTmarked H. sapiens K562 cells P281 NPAT   1,619,203
SH_Hs_NPB4_20190217.DTmarked H. sapiens K562 cells P281 NPAT   1,122,716
SH_Hs_NPB8_20190217.DTmarked H. sapiens K562 cells P281 NPAT   640,055

| Antibodies | anti-RNAPII (mouse) Covance Research Products MMS-126R |
|---|---|
| | anti-RNAPII-S5p (rabbit) Cell Signalling Technology D9N5I |
| | anti-RNAPII-S2p (rabbit) Cell Signalling Technology E1Z3G |
| | anti-Mxc (guinea pig) RJ Duronio Mxc-C-1 |
| | anti-Mute (guinea pig) RJ Duronio Mute |
| | anti-H3K9me1 (rabbit) Epicypher 13-0029 |
| | anti-H3K9me2 (mouse) EMD Millipore 05-1249 |
| | anti-H3K9me3 (rabbit) Abcam ab8898 |
| | anti-H3K27me3 (rabbit) Cell Signalling Technology C36B11 |
| | anti-uH2A (rabbit) Cell Signalling Technology D27C4 |
| | anti-H3K4me1 (rabbit) Epicypher 13-0057 |
| | anti-H3K4me2 (rabbit) Epicypher 13-0027 |
| | anti-H3K4me3 (rabbit) Epicypher 13-0060 |
| | anti-H3K27ac (rabbit) Epicypher 13-0059 |
| | anti-H3K36me3 (rabbit) Epicypher 13-0058 |
| | anti-MPM2 (mouse) DAKO M3514 |
| | anti-GFP (rabbit) Cell Signalling Technology D5.1 |
| | anti-GFP (mouse) Thermo Fisher Scientific 3E6 |
| | anti-rabbit IgG (guinea pig) Antibodies Online ABIN101961 |
| | anti-mouse IgG (rabbit) Abcam ab46540 |
| | anti-guinea pig IgG (rabbit) Thermo Fisher Scientific PA1-28549 |
| | anti-histone H4 (mouse) Abcam ab31830 |
| | anti-NPAT (rabbit) Thermo Fisher Scientific PA565419 |

| Peak calling parameters | No peaks were called. |
|---|---|
| Data quality | Quality was assessed by correlation between replicates. |
| Software | bamtools, featureCounts |

# Flow Cytometry

## Plots

Confirm that:

☐ The axis labels state the marker and fluorochrome used (e.g. CD4-FITC).

☐ The axis scales are clearly visible. Include numbers along axes only for bottom left plot of group (a 'group' is an analysis of identical markers).

☐ All plots are contour plots with outliers or pseudocolor plots.

☐ A numerical value for number of cells or percentage (with statistics) is provided.

## Methodology

| | |
|---|---|
| Sample preparation | *Describe the sample preparation, detailing the biological source of the cells and any tissue processing steps used.* |
| Instrument | *Identify the instrument used for data collection, specifying make and model number.* |
| Software | *Describe the software used to collect and analyze the flow cytometry data. For custom code that has been deposited into a community repository, provide accession details.* |
| Cell population abundance | *Describe the abundance of the relevant cell populations within post-sort fractions, providing details on the purity of the samples and how it was determined.* |
| Gating strategy | *Describe the gating strategy used for all relevant experiments, specifying the preliminary FSC/SSC gates of the starting cell population, indicating where boundaries between "positive" and "negative" staining cell populations are defined.* |

☐ Tick this box to confirm that a figure exemplifying the gating strategy is provided in the Supplementary Information.

# Magnetic resonance imaging

## Experimental design

| | |
|---|---|
| Design type | *Indicate task or resting state; event-related or block design.* |
| Design specifications | *Specify the number of blocks, trials or experimental units per session and/or subject, and specify the length of each trial or block (if trials are blocked) and interval between trials.* |
| Behavioral performance measures | *State number and/or type of variables recorded (e.g. correct button press, response time) and what statistics were used to establish that the subjects were performing the task as expected (e.g. mean, range, and/or standard deviation across subjects).* |

## Acquisition

| | |
|---|---|
| Imaging type(s) | *Specify: functional, structural, diffusion, perfusion.* |
| Field strength | *Specify in Tesla* |
| Sequence & imaging parameters | *Specify the pulse sequence type (gradient echo, spin echo, etc.), imaging type (EPI, spiral, etc.), field of view, matrix size, slice thickness, orientation and TE/TR/flip angle.* |
| Area of acquisition | *State whether a whole brain scan was used OR define the area of acquisition, describing how the region was determined.* |

Diffusion MRI      ☐ Used      ☐ Not used

## Preprocessing

| | |
|---|---|
| Preprocessing software | *Provide detail on software version and revision number and on specific parameters (model/functions, brain extraction, segmentation, smoothing kernel size, etc.).* |
| Normalization | *If data were normalized/standardized, describe the approach(es): specify linear or non-linear and define image types used for transformation OR indicate that data were not normalized and explain rationale for lack of normalization.* |
| Normalization template | *Describe the template used for normalization/transformation, specifying subject space or group standardized space (e.g. original Talairach, MNI305, ICBM152) OR indicate that the data were not normalized.* |
| Noise and artifact removal | *Describe your procedure(s) for artifact and structured noise removal, specifying motion parameters, tissue signals and physiological signals (heart rate, respiration).* |
| Volume censoring | *Define your software and/or method and criteria for volume censoring, and state the extent of such censoring.* |

## Statistical modeling & inference

| | |
|---|---|
| Model type and settings | *Specify type (mass univariate, multivariate, RSA, predictive, etc.) and describe essential details of the model at the first and second levels (e.g. fixed, random or mixed effects; drift or auto-correlation).* |
| Effect(s) tested | *Define precise effect in terms of the task or stimulus conditions instead of psychological concepts and indicate whether ANOVA or factorial designs were used.* |

Specify type of analysis:      ☐ Whole brain      ☐ ROI-based      ☐ Both

Statistic type for inference

(See Eklund et al. 2016)

> *Specify voxel-wise or cluster-wise and report all relevant parameters for cluster-wise methods.*

Correction

> *Describe the type of correction and how it is obtained for multiple comparisons (e.g. FWE, FDR, permutation or Monte Carlo).*

## Models & analysis

| n/a | Involved in the study |
|-----|----------------------|
| ☐ ☐ | Functional and/or effective connectivity |
| ☐ ☐ | Graph analysis |
| ☐ ☐ | Multivariate modeling or predictive analysis |

Functional and/or effective connectivity

> *Report the measures of dependence used and the model details (e.g. Pearson correlation, partial correlation, mutual information).*

Graph analysis

> *Report the dependent variable and connectivity measure, specifying weighted graph or binarized graph, subject- or group-level, and the global and/or node summaries used (e.g. clustering coefficient, efficiency, etc.).*

Multivariate modeling and predictive analysis

> *Specify independent variables, features extraction and dimension reduction, model, training and evaluation metrics.*

