## [Peer Review File · Nature Structural & Molecular Biology]

Cell-cycle-dependent repression of histone gene transcription by histone H4

Corresponding Author: Dr Steven Henikoff

Version 0:

Decision Letter:

2nd Jul 2025

Dear Dr Henikoff,

Thank you again for submitting your manuscript "Cell-cycle-dependent repression of histone gene transcription by histone H4". I apologise for the delay in responding, which resulted from the difficulty in timely obtaining and discussing suitable referee reports. Nevertheless, we now have comments (below) from the 3 reviewers who evaluated your paper. In light of those reports, we remain interested in your study and would like to see your response to the comments of the referees, in the form of a revised manuscript.

You will see that though all reviewers acknowledge and appreciate the potential of the study, if it is technically solidified, they raise fairly similar concerns about issues limiting the confidence in the proposed model and lack of mechanistic clarity. While we agree that mechanistic insight into how non-nucleosomal H4 would exert its purported repressive roles would be amazing to obtain, we editorially deem that it is not a prerequisite for the success of this story. However, we fully agree with the reviewers that all experimental concerns, including but not limited to lack of data quantification, missing controls, lack of orthogonal technical validation with additional antibodies as requested by Reviewer #2, need for better figure presentation and annotation, must be comprehensively and convincingly addressed in a revised manuscript for the editorial team to be able to re-engage with the referees and to increase the chances of success of the manuscript.

In addition to answering these issues, we editorially agree with the proposal of all referees that the study would benefit from further H4 depletion experiments in both flies and human cells (we editorially note that if the depletion was targeted to the non-nucleosomal part of H4 it would provide a huge boost to these experiments) and H4 overexpression experiments in flies, amongst other proposed ways to acquire further support for the proposed model.

Please be sure to address/respond to all concerns of the referees in full in a point-by-point response and highlight all changes in the revised manuscript text file.

We appreciate the requested revisions could require an extended deadline to fully address, we thus expect to see your revised manuscript within 3-6 months. If you cannot send it within this time, please let us know. We will be happy to consider your revision as long as nothing similar has been accepted for publication at NSMB or published elsewhere. Should your manuscript be substantially delayed without notifying us in advance and your article is eventually published, the received date would be that of the revised, not the original, version.

Reporting Summary:

When submitting the revised version of your manuscript, please pay close attention to our <https://www.nature.com/nature-portfolio/editorial-policies/image-integrity> Digital Image Integrity Guidelines and

to the following points below:

- that unprocessed scans are clearly labelled and match the gels and western blots presented in figures. Please note that all key data shown in the main figures as cropped gels or blots should be presented in uncropped form, with molecular weight markers. While these data can be displayed in a relatively informal style, they must refer back to the relevant figures. These data should be submitted as source data with the last revision, prior to acceptance.
- that control panels for gels and western blots are appropriately described as loading on sample processing controls
- all images in the paper are checked for duplication of panels and for splicing of gel lanes.
- For any revision that includes light microscopy data, we ask our authors to please include a completed light microscopy reporting table (https://www.nature.com/documents/Light_microscopy_reporting_table.xlsx) to ensure the methods are described thoroughly. The table will be available to reviewers and ultimately published should the manuscript be accepted at the journal.

EXTENDED DATA FIGURES

We require deposition of coordinates (and, in the case of crystal structures, structure factors) into the Protein Data Bank with the designation of immediate release upon publication (HPUB). Electron microscopy-derived density maps and coordinate data must be deposited in EMDB and released upon publication. Deposition and immediate release of NMR chemical shift assignments are highly encouraged. Deposition of deep sequencing and microarray data is mandatory, and the datasets must be released prior to or upon publication. To avoid delays in publication, dataset accession numbers must be supplied with the final accepted manuscript and appropriate release dates must be indicated at the galley proof stage. Please find the complete NRG policies on data availability at <http://www.nature.com/authors/policies/availability.html>.

Link Redacted

Sincerely,

Dimitris Typas

Reviewers' Comments:

Reviewer #1 (Remarks to the Author):

In this manuscript by Ahmed et al., the authors set off to understand why some of histone genes in the highly repeated histone locus are repressed why others are transcriptionally active. By profiling several known factors that are associated with activation and/or repression of histone gene expression, they establish a quantitative framework that shows which marks are enriched or depleted in the full-length array vs. a reduced array that has only 12 histone repeats. This provides evidence that histone repeats can either be active or silenced in the context of the full array or that all the histone genes are active at an intermediate level. More importantly, the authors use a reporter system and provide data that supports the idea that histone genes can be active or silenced depending on the number of histone repeats. Then, they establish the *Drosophila testis* as system to visualize histone gene expression in the germline of living animals. Here they show that depletion of only His4 causes depression of His2A and His3 reporters. They also show that His4 localizes to the HLB and that the presence of His4 is correlated with reduced histone expression (RNAPII-S5p). They also demonstrate that His4 localized to the HLB in human cells.

This manuscript is very well written, and the data is of high quality. The authors are careful not to make conclusions beyond what their data support. While some of the early sections of this paper are largely confirmatory, there are two important aspects to this work. First, it established the *Drosophila testis* as a germline model to understand the regulation of histone gene expression. Second, it provides the first evidence that histone 4 acts as a regulator of gene expression independent of the nucleosome. Furthermore, this function of histone 4 is conserved and could represent an early function of histone H4 in evolution. The weakness of this work is that it doesn't provide any mechanistic insight into how H4 represses histone gene expression. Regardless, this will be a paradigm-shifting paper in our understanding of histone biosynthesis and those mechanistic details will emerge over many years.

While I have some suggestions to make the conclusions more robust and at times more quantitative, my concerns are largely minor.

1. Could 12X produce fewer histones, thus resulting in fewer nucleosomes incorporated into chromatin and alter transcription throughout the genome? While the H3-Dendra reporter is promising, it would be nice to know that this is specific for histones and not a general effect on transcription genome wide in 12X imaginal discs. Same for the germ cells, what happens the the His2AV reporter in the 12X background? I would assume there is no change but that would be nice to confirm.

2. Does KD of H4 affect the H2AV reporter?

3. It would be helpful to quantify the derepression of the fluorescent reports in Figure 4. His2A looks fairly obvious but His3A does not. Were these images taken with the same exposure etc.? The RFP looks brighter for the H4 KD samples so I'm not sure how to compare.

4. The knockdown data for 4I is very interesting. Unfortunately, there is no quantification of knock down levels so it's nearly impossible to conclude anything from a no phenotype RNAi experiment. The authors should at least acknowledge this possible interpretation

Similarly, there should be some sort of quantification for the RNAPII-S5p staining in Figure 7. The claim is that the signal is reduced in G2 cells and this is dependent on H4. Quantification of the intensities +/- H4 would show this more clearly.

5. It seems that the authors could do a little more to test their model. For example, what happens to histone gene expression when H4 is overexpressed using the UAS-His4 line?

Minor text and figure issues?

Line 119: In contrast

Figure 2: I can't figure out why some text values are black, and some are gray.

I don't know what the color code is for Figure 8D. I would be nice to have a scale/label on the figure. Does darker mean more signal of that mark?

The last paragraph of the discussion feels very disconnected from the rest of the story. I understand that they want to appeal to broad audience, but they may want to consider revising this to make the flow more logical (just my opinion).

Reviewer #2 (Remarks to the Author):

Ahmad et al present interesting results regarding how histone gene expression is controlled, uncovering a potentially novel form of negative feedback regulation that helps maintain the tight coupling between histone protein biosynthesis and DNA replication. Using genomic and cell biological readouts the authors present data in *Drosophila* and human cells that histone H4 accumulates in the Histone Locus Body (HLB), a nuclear body that associates with the replication-coupled histone genes and contains factors necessary for histone mRNA production. Using fluorescent reporters for histone gene expression and genetic manipulation of chromatin regulators in *Drosophila* testes, the authors argue that histone H4 accumulation in the HLB provides a form of negative feedback regulation on histone transcription that functions to down regulate histone gene expression when the level of free histones outpace incorporation into nucleosomes, as would happen at the end of S phase/early G2 when genome duplication is complete. Although this model is incredibly attractive, the data supporting it are not extensive, at times not at all clear (e.g. Fig. 5e-g; see later paragraph), and do not address the molecular mechanism of the negative feedback regulation proposed by the authors.

Although determining the details of a possibly complex negative feedback mechanism involving H4 localization to the HLB is beyond the scope of a single paper, the absence of any substantial discussion of the matter is conspicuous. There is no discussion about what the anti H4 mouse monoclonal antibody from Abcam might be recognizing. At face value it seems clear that the antibody is not recognizing all the H4 in the nucleus or the cell, but rather a special sub population of H4 molecules that is enriched in the HLB. What might be special about the epitope that this antibody is recognizing, and thus the pool of H4 that is localized to the HLB? Presumably it is not assembled into a nucleosome and is thus "free". But is the H4 also not bound to H3 or a chaperone complex thereby having an exposed epitope that the antibody can recognize, but not when H4 is assembled into a nucleosome? Might it still be bound to H3 in the HLB but the available H3 antibodies cannot recognize the pool of H4-bound H3 in the HLB? How might "free" H4 get to the HLB? Do other H4 antibodies recognize this pool?

The most important, and novel, data pertaining to the mechanism of the H4-driven negative feedback model, which is supported by the increase in histone gene reporter activity after H4 knockdown in *Drosophila*, is the accumulation of endogenous H4 in the HLB. However, the *Drosophila* data in this regard shown in Fig. 5e-g is problematic. What each color in the figure represents is unclear. The legend for panel e says that H4 is indicated in green, but in the figure the H4 is indicated in red. Moreover, the split out H4 panels in e and f do not look like the green signal in the merge (which isn't labeled in the figure); is that panel maybe Mxc rather than H4? Confusion like this reduces overall confidence in this key data, even if the H4GFP ectopic expression experiments demonstrate that ectopically expressed H4 CAN accumulate in the HLB. Also, knocking down H4 in both flies and human cells to ask whether that manipulation reduces or eliminates the HLB H4 signal is an important experiment.

There are no controls for the degree of knockdown via RNAi in flies. Thus, even implying that "cells measure the demand for histones based only on the H4 histone" (line 220) is too strong for the data shown.

Much of the data are not quantified, particularly the signals being measured in HLBs after manipulation of H4 in Figure 7 and elsewhere. Such measurements are critical for readers to rigorously interpret the data and evaluate the conclusions made by the authors. Another specific example: there is insufficient description of how a 37x fold increase of *Dendra2* expression in the 12XWT background was measured.

The presentation of cell biological data overall should be improved. Not all figures have scale bars, some of the images could be better labeled with genotypes and arrows pointing out key features, and often not a sufficiently high enough level of magnification is used to interpret the images. Specific examples are provided in the Figure-by-Figure comments below.

Figure 1

For all experiments it is not entirely clear exactly how many histone genes were in the cells that were analyzed; i.e. 12x or 24x, 100x or 200x? More clarity on this issue will help the reader think about how the data were normalized for presentation.

Was Dm6 used for Figure 11, or was a single histone repeat unit that aggregates all the reads used? If the former, did the authors just choose a single HRU from the data in panel a, and is this representative? This question is important to interpret that shape of the cut and tag signal over the individual histone genes.

Can the authors comment on why there is no (or very low) Mute signal on the H1 gene, especially considering the recent papers on CRAMP1 (which did come out after submission). PMIDs: 40516529, 40516528, 39829857.

Figure 2

More detail about the features of this table would be helpful. e.g. For each mark how were cutoffs for biological significance determined and why? What numbers in panel a are "important"? What is difference between grey font and black font? What does the green heat map represent?

Figure 3

What is the exact genotype of the tissues. Most important, is "wildtype" a 200X or 100X because it's a sibling of the 12XWT. And how was the average summed fluorescence determined? Field of view (the entire disc or a portion) and how many discs?

Figure 4

The DAPI staining is strange; it doesn't appear if nuclei are being labeled.

Carrying the dashed lines over to the color panels would be helpful.

Need to point to the sheath cells with an arrow and show a high mag view; they are nearly impossible to see as is.

A high mag image for panel h like in panels c, e, g would be helpful.

Panel g doesn't show a sheath cell as the legend indicates; those look like the gonial cells.

The bottom line is that without more guidance the impact will be far lower for non *Drosophila* readers without this help.

Figure 4

They don't show any KD efficiencies for any of the target

What is the asterisk in panel i? What is the red/green cutoff based on? A quantifiable measurement or just inspection?

They don't formally show a control for the RNAi knockdowns

Scale bars on all the figures and the actual scale is needed.

Figure 5

Discussed above.

Figure 6 and 7

There is no quantification of signals in any HLBs; measuring the intensity of MPM2 and S5p in the HLBs would increase the rigor of the conclusion that H4 KD increase S5p signal.

Figure 8

Panel of heat map is blurry and needs to be improved.

Reviewer #3 (Remarks to the Author):

The manuscript by Ahmad and colleagues uncovers a novel regulatory mechanism that couples DNA synthesis and histone deposition whereby cells prevent excess histone synthesis by histone H4 binding to the promoters of canonical histone genes. Using an elegant cellular system in fly, and molecular and cellular approaches, they demonstrate that histone H4 is specifically targeted to Histone Locus bodies (HLB), where it binds to promoters, and that the level of H4 negatively correlates with histone gene expression and HLB activity. Previously, they characterized the molecular requirements that define active histone unit repeats by comparing wild type genotype with a strain with a minimal (and highly active) set of histone copies. The text is clear, evidence is solid and, importantly, the conclusions are highly relevant, supporting a novel, simple and apparently conserved model to prevent the genotoxic effects of unbalanced histone synthesis.

Several points should be addressed before its publication.

1. An interesting aspect of the work is the conservation of the mechanism in human cells. Data clearly show that histone H4 binds to promoters of NPAT-transcriptionally activated histone genes. As indicated by the authors, this implicates histone H4 in the S phase regulation of canonical histone gene promoters in human cells. However, authors should demonstrate that partial histone H4 depletion also causes histone derepression in human cells. The fact that both H4 and NPAT coincides at promoters suggests that the expression level is modulated and could be potentially increased (after H4 reduction) or decreased (after NPAT reduction).

2. Figures 4j-m and 7c-f: Even though the effect displayed in the representative images are clear, a quantification of the effects would be advisable.

3. Some controls should be included in the following panels:

- 4h: RFP and merge (they are modified testis and gonial cells should be shown)

- 5h or 5i: Histone H3 control; especially because a specific increase in H3K27ac is observed. Same for Figure 8 (either a or b), especially taking into account that authors try to demonstrate that the mechanism is conserved.

Minor points:

- Fig 2: Please list 2a and 2b in the same order for clarity with the text.

- Lines 143/144: the distribution of epigenetic marks does not allow to rule out the possibility that, rather than a mixture of active and inactive copies, as in the ribosomal locus, histones copies in the array are all partially expressed.

- Fig 7 text: c,e is c,d; and d,f is e,f.

Version 1:

Decision Letter:

Our ref: NSMB-A51044A

27th Oct 2025

Dear Dr. Henikoff,

Thank you for submitting your revised manuscript "Cell-cycle-dependent repression of histone gene transcription by histone H4" (NSMB-A51044A). It has now been seen by the original referees and their comments are below. The reviewers find that the paper has improved in revision, and therefore we are happy to accept it in principle in Nature Structural & Molecular Biology, pending minor revisions to satisfy the referees' final requests, such as the remaining points of Reviewer #2 on adding certain citations, better presentation of EdU in a certain panel and of the H4 KD in the HLB of testes in Supplemental Figure 2, and to comply with our editorial and formatting guidelines.

We are now performing detailed checks on your paper and will send you a checklist detailing our editorial and formatting requirements in about two weeks. Please do not upload the final materials and make any revisions until you receive this additional information from us.

Sincerely,

Dimitris Typas
Senior Editor
Nature Structural & Molecular Biology
ORCID: 0000-0002-8737-1319

Reviewer #1 (Remarks to the Author):

The authors addressed all the points I raised and I am in full support for this article to be published in NSMB. The complementary manuscript on H4 arginine-3 di-methylation further validates the importance of the current manuscript.

Reviewer #2 (Remarks to the Author):

The authors made an excellent effort to address all the reviewer comments, and the manuscript is clearer as a result. I also feel that the Discussion is improved and better tied to the results presented. I only have a few remaining thoughts and simple suggestions below.

Could also cite PMID: 40516529 and PMID: 21336627 on line 85.

Could also cite PMID: 40397569 on line 94 and lines 249-253 where the results are similar to those in wing discs, and line 303.

The conclusion made in lines 111-112 was also made in PMID: 25669886, which could be cited here.

In Figure 5f you might show the pink EdU channel separately, as you did in panel e, to make it abundantly clear that that nucleus is not in S phase like the one in panel e.

The knockdown of H4 in the HLB of testes in Supplemental Figure 2 is not convincingly shown (or very understandable), and there is no figure legend to help understand the data (e.g. what is the arrowhead pointing out?). A better presentation of these data is needed for them to be interpreted appropriately.

I still think the authors should use total histone gene count (i.e. 200x and 24x) per diploid genome rather than 100x and 12x haploid "copies/genome" number; it just seems more intuitive to me (this is a very minor point).

Reviewer #3 (Remarks to the Author):

All my concerns have been properly addressed, and the changes introduced in response to reviewers' critical points have successfully improved the manuscript, which provides a simple and attractive mechanism to understand how histone

synthesis is coupled to histone demand.

Version 2:

Decision Letter:

20th Nov 2025

Dear Dr. Henikoff,

We are now happy to accept your revised paper "Cell-cycle-dependent repression of histone gene transcription by histone H4" for publication as an Article in Nature Structural & Molecular Biology.

Your paper will be published online soon after we receive proof corrections and will appear in print in the next available issue. You can find out your date of online publication by contacting the production team shortly after sending your proof corrections.

An online order form for reprints of your paper is available at <https://www.nature.com/reprints/author-reprints.html>. Please let your coauthors and your institutions'

public affairs office know that they are also welcome to order reprints by this method.

Authors may need to take specific actions to achieve compliance with funder and institutional open access mandates. If your research is supported by a funder that requires immediate open access (e.g. according to [Plan S principles](https://www.springernature.com/gp/open-science/plan-s-compliance) or the [NIH public access policy](https://www.springernature.com/gp/open-science/us-federal-agency-compliance)) then you should select the gold OA route, and we will direct you to the compliant route where possible. Because authors warrant under our subscription licensing terms that they haven't committed to licensing any version of their article under a licence inconsistent with the terms of our agreement – including the applicable embargo period – publication under the subscription model isn't suitable for authors whose funders require no embargo.

Sincerely,

Dimitris Typas
Senior Editor
Nature Structural & Molecular Biology
ORCID: 0000-0002-8737-1319

Reviewers' Comments:

We thank all three reviewers for their overall enthusiasm for our study, and for their many perceptive comments and suggestions that have greatly improved the manuscript.

Reviewer #1 (Remarks to the Author):

In this manuscript by Ahmed et al., the authors set off to understand why some of histone genes in the highly repeated histone locus are repressed why others are transcriptionally active. By profiling several known factors that are associated with activation and/or repression of histone gene expression, they establish a quantitative framework that shows which marks are enriched or depleted in the full-length array vs. a reduced array that has only 12 histone repeats. This provides evidence that histone repeats are can either be active or silenced in the context of the full array or that all the histone genes are active at an intermediate level. More importantly, the authors use a reporter system and provide data that supports the idea that histone genes can be active or silenced depending on the number of histone repeats. Then, they establish the *Drosophila testis* as system to visualize histone gene expression in the germline of living animals. Here they show that depletion of only His4 causes depression of His2A and His3 reporters. They also show that His4 localizes to the HLB and that the presence of His4 is correlated with reduced histone expression (RNAPII-S5p). They also demonstrate that His4 localized to the HLB in human cells.

This manuscript is very well written, and the data is of high quality. The authors are careful not to make conclusions beyond what their data support. While some of the early sections of this paper are largely confirmatory, there are two important aspects to this work. First, it established the *Drosophila testis* as a germline model to understand the regulation of histone gene expression. Second, it provides the first evidence that histone 4 acts as a regulator of gene expression independent of the nucleosome. Furthermore, this function of histone 4 is conserved and could represent an early function of histone H4 in evolution. The weakness of this work is that it doesn't provide any mechanistic insight into how H4 represses histone gene expression. Regardless, this will be a paradigm-shifting paper in our understanding of histone biosynthesis and those mechanistic details will emerge over many years.

While I have some suggestions to make the conclusions more robust and at times more quantitative, my concerns are largely minor.

1. Could 12X produce fewer histones, thus resulting in fewer nucleosomes incorporated into chromatin and alter transcription throughout the genome? While the H3-Dendra reporter is promising, it would be nice to know that this is specific for histones and not a general effect on transcription genome wide in 12X imaginal discs.

We have added new analyses to address this question. We used data profiling RNAPII in wildtype and 12X wing discs to measure gene transcription genome-wide. Differential analysis does not detect significant differences in gene expression between these genotypes, including total histone gene expression. This confirms published genetic

results that the 12X array is sufficient. We have added this analysis in Supplementary Figure 1, Supplementary Table 3, and in the text at line 106-109.

Same for the germ cells, what happens the His2AV reporter in the 12X background? I would assume there is no change but that would be nice to confirm.

We performed this experiment; there is no change. We have added this data to Figure 4 and describe the result on lines 191-193.

2. Does KD of H4 affect the H2AV reporter?

We performed this experiment; there is no change. This data is added to Figure 4l and described on lines 219-221.

3. It would be helpful to quantify the derepression of the fluorescent reports in Figure 4. His2A looks fairly obvious but His3A does not. Were these images taken with the same exposure etc.? The RFP looks brighter for the H4 KD samples so I'm not sure how to compare.

We have added quantification of the fluorescence intensity in all genotypes. This is now presented in Figure 4q and discussed on lines 216-218.

4. The knockdown data for 4l is very interesting. Unfortunately, there is no quantification of knock down levels so it's nearly impossible to conclude anything from a no phenotype RNAi experiment. The authors should at least acknowledge this possible interpretation.

We agree with the reviewer that other negative results in the testis does not - on their own - rule out a role for another histone in negative regulation. However, the lack of cytological signal for H3 variants in the HLB also argues that these histones have no special role in the HLB. We have emphasized in the text the limitations of negative results in this screen on line 206-208.

Similarly, there should be some sort of quantification for the RNAPII-S5p staining in Figure 7. The claim is that the signal is reduced in G2 cells and this is dependent on H4. Quantification of the intensities +/- H4 would show this more clearly.

We have quantified the staining for RNAPIIS5p and MPM2 in the HLB of these two genotypes in the testis, and now present the quantitation for RNAPIIS5p in the text in lines 264-271 and in Figure 7g.

5. It seems that the authors could do a little more to test their model. For example, what happens to histone gene expression when H4 is overexpressed using the UAS-His4 line?

We have attempted to overexpress histone H4 in Drosophila, but our existing constructs produce only trace amounts of tagged histone. However, our discussion of this work at recent conferences in particular inspired a second group to revisit some cryptic results on inhibitors of the PRMT5 arginine di-methyltransferase. This is one of the most promising epigenetic anti-cancer drug targets, but its function has been mysterious. This group has gone on to confirm our results on HLB localization of histone H4, and to demonstrate that PRMT5-mediated histone H4 arginine-3 di-methylation disrupts this localization. This group's manuscript is now in review, and we highlight some relevant

details from their preprint (<https://www.biorxiv.org/content/10.1101/2025.07.03.663002v1>) below, and in the Discussion of our manuscript. The importance of this pathway is emphasized by their demonstration that the primary effect of anti-cancer PRMT5 inhibitors is to down-regulate histone gene expression, and not through affecting mRNA splicing has been previously believed. We believe that our two papers together stimulate substantial interest in this area.

Minor text and figure issues?

Line 119: In contrast

Fixed.

Figure 2: I can't figure out why some text values are black, and some are gray.

All reviewers had difficulty interpreting this table, so we improved its presentation, showing only those modifications that have measurable enrichment at histone genes (in the previous version gray values indicated histone modification signals at background levels, thus fold-changes that had little meaning). We state in the text that we focused on modifications above background (lines 136-137).

I don't know what the color code is for Figure 8D. I would be nice to have a scale/label on the figure. Does darker mean more signal of that mark?

We added scales for each mark, indicating that darker colors represent more signal counts. Counts for this table are provided in Supplementary Table 2.

The last paragraph of the discussion feels very disconnected from the rest of the story. I understand that they want to appeal to broad audience, but they may want to consider revising this to make the flow more logical (just my opinion).

We added text (lines 339-359) to describe recent work that provides independent support for our points in this paragraph. The recent observation that anti-cancer inhibitors of the PRMT5 arginine di-methyltransferase act on soluble histone H4 in HLBs and rapidly block histone gene expression strongly supports the connections we make between cancer and histone autoregulation.

Reviewer #2 (Remarks to the Author):

Ahmad et al present interesting results regarding how histone gene expression is controlled, uncovering a potentially novel form of negative feedback regulation that helps maintain the tight coupling between histone protein biosynthesis and DNA replication. Using genomic and cell biological readouts the authors present data in *Drosophila* and human cells that histone H4 accumulates in the Histone Locus Body (HLB), a nuclear body that associates with the replication-coupled histone genes and contains factors necessary for histone mRNA production. Using fluorescent reporters for histone gene expression and genetic manipulation of chromatin

regulators in *Drosophila* testes, the authors argue that histone H4 accumulation in the HLB provides a form of negative feedback regulation on histone transcription that functions to down regulate histone gene expression when the level of free histones outpace incorporation into nucleosomes, as would happen at the end of S phase/early G2 when genome duplication is complete. Although this model is incredibly attractive, the data supporting it are not extensive, at times not at all clear (e.g. Fig. 5e-g; see later paragraph), and do not address the molecular mechanism of the negative feedback regulation proposed by the authors.

We address these concerns with additional experiments, improved figures, further quantitative analysis of images, text changes for clarification, and important new results from another group, described below.

Although determining the details of a possibly complex negative feedback mechanism involving H4 localization to the HLB is beyond the scope of a single paper, the absence of any substantial discussion of the matter is conspicuous.

We have expanded our comments on this in the Discussion, including new comments on a recent preprint that cites our preprint and acknowledges our discussions with them as inspiration for their work (<https://www.biorxiv.org/content/10.1101/2025.07.03.663002v1>) (lines 339-359). This group was studying the arginine symmetrical di-methyltransferase, PRMT5, one of the most promising epigenetic anti-cancer drug targets, but its function has been mysterious. They went on to confirm our results on HLB localization of histone H4, and to demonstrate that PRMT5-mediated histone H4 arginine-3 di-methylation disrupts this localization. The importance of this pathway is emphasized by their demonstration that the primary effect of anti-cancer PRMT5 inhibitors is to down-regulate histone gene expression, and not through affecting mRNA splicing has been previously believed. We believe that our two papers together stimulate substantial interest in this area.

There is no discussion about what the anti H4 mouse monoclonal antibody from Abcam might be recognizing. At face value it seems clear that the antibody is not recognizing all the H4 in the nucleus or the cell, but rather a special sub population of H4 molecules that is enriched in the HLB. What might be special about the epitope that this antibody is recognizing, and thus the pool of H4 that is localized to the HLB? Presumably it is not assembled into a nucleosome and is thus “free”. But is the H4 also not bound to H3 or a chaperone complex thereby having an exposed epitope that the antibody can recognize, but not when H4 is assembled into a nucleosome?

We have added comments to the text describing what we think is going on with this antibody (lines 236-239). We interpret our results just as the reviewer is suggesting. The antibody was raised to unmodified histone H4. Our staining patterns and genomic profiling patterns in both *Drosophila* and in human cells indicate that it does not recognize nucleosomal histone H4 that is broadly distributed. Inclusion in a nucleosome occludes the histone fold domain and greatly reduces access to the n-terminal tail of H4 (PMID: 30293810). The antibody has been used primarily on Western blots with denatured protein. Thus, it is plausible that this antibody only recognizes non-nucleosomal histone H4.

Might it still be bound to H3 in the HLB but the available H3 antibodies cannot recognize the pool of H4-bound H3 in the HLB?

We argue that this is not the case, because H3-GFP fusion proteins are not affected by epitope occlusion and these do not label the HLB. This data is presented in Figure 5.

How might “free” H4 get to the HLB?

This is a fascinating question, and we now provide a possible molecular mechanism inspired by a very recent preprint (Abidi AA, Dailey GM, Tjian, R, Thomas Graham GW (2025) Collective unstructured interactions drive chromatin binding of transcription factors bioRxiv doi: 10.1101/2025.05.16.654615):

“Mathematical modeling has suggested that feedback from soluble histone pools is necessary for precise coupling between DNA replication and histone synthesis⁵⁴. Our observations suggest a simple model where soluble histone H4 protein directly represses histone gene transcription. Ongoing DNA replication and chromatin packaging use up soluble histones, but once DNA replication ceases, soluble histones including histone H4 accumulate. Monomeric histones have been observed in cells³⁸, and so a monomeric histone H4 might directly interact with NPAT at histone gene promoters. The majority of the NPAT and Mxc proteins are composed of intrinsically-disordered domains (IDRs) interspersed with structured domains, and both are important for self-associations and function^{55,56}. We propose an interaction between the strong positive charge of the unmodified H4 N-terminal tail (N-SGRGKGGKGLGKGGAKRHRKVLRL) and the strong negative charge of the unstructured regions of phosphorylated NPAT. Weak, transient interactions have been observed to drive chromatin binding of transcription factors⁵⁷, and in the case of H4-NPAT would make H4R3 available for access by PRMT5 for arginine di-methylation and H4 release from NPAT.” (lines 307-320).

Do other H4 antibodies recognize this pool?

We note that an independent antibody to histone H4 also labels the HLB (<https://www.biorxiv.org/content/10.1101/2025.07.03.663002v1>); this together with our results with histone H4-GFP localization to the HLB confirms the result.

The most important, and novel, data pertaining to the mechanism of the H4-driven negative feedback model, which is supported by the increase in histone gene reporter activity after H4 knockdown in *Drosophila*, is the accumulation of endogenous H4 in the HLB. However, the *Drosophila* data in this regard shown in Fig. 5e-g is problematic. What each color in the figure represents is unclear. The legend for panel e says that H4 is indicated in green, but in the figure the H4 is indicated in red. Moreover, the split out H4 panels in e and f do not look like the green signal in the merge (which isn't labeled in the figure); is that panel maybe Mxc rather than H4? **Fixed; For Figure 5e the figure labeling was correct but not the legend. In panels 5e and 5f we are using antibody to H4 which we label in red; green in Fig 5e,f is DAPI staining of the nucleus. These samples are not labeled for Mxc.**

Confusion like this reduces overall confidence in this key data, even if the H4GFP ectopic expression experiments demonstrate that ectopically expressed H4 CAN accumulate in the HLB.

We agree this is key data, where the antibody shows that endogenous histone H4 is in the HLB. Staining and chromatin profiling with this antibody is shown in Figures 5e,f,h, and i for *Drosophila*, and in Figures 8a-d for human cells, demonstrating in multiple experiments that endogenous histone H4 is localized to the HLB.

Also, knocking down H4 in both flies and human cells to ask whether that manipulation reduces or eliminates the HLB H4 signal is an important experiment.

We performed this experiment in flies, and knockdown of histone H4 greatly reduces H4-GFP in the HLB. We now present the result in Supplementary Figure 2 and mention it in the text on lines 269-270. As for human cells, we have a much better way of addressing this issue, by referring to the preprint described above

(<https://www.biorxiv.org/content/10.1101/2025.07.03.663002v1>). In this cytological study, the authors showed that H4R3 dimethylation releases H4 from NPAT, and that within 15 minutes of PRMT5 inhibition, histone gene transcript levels were reduced, which implies that the PRMT5 H4R3 dimethylation directly interferes with H4-NPAT binding.

There are no controls for the degree of knockdown via RNAi in flies. Thus, even implying that “cells measure the demand for histones based only on the H4 histone” (line 220) is too strong for the data shown.

We have added experiments assessing knockdown efficiency for *His2A* and *His4* as Supplementary Figure 2, where we have reporters to test it. We have expanded our comment pointing out the limitations of negative RNAi results in the text (line 206-208 and lines 213-216). We emphasize that we are basing our interpretation on the positive evidence we have for a role for histone H4 in histone gene repression.

Much of the data are not quantified, particularly the signals being measured in HLBs after manipulation of H4 in Figure 7 and elsewhere. Such measurements are critical for readers to rigorously interpret the data and evaluate the conclusions made by the authors.

We agree with the reviewer, and have added image quantitation of images for Figure 4 and Figure 7. We refer to these measurements on lines 216-218 and 264-271, respectively.

Another specific example: there is insufficient description of how a 37x fold increase of *Dendra2* expression in the 12XWT background was measured.

We have added details to the *Methods/Imaging fresh tissues* section (lines 381-385), and summarize it briefly in the legend to Figure 3b (lines 698-700).

The presentation of cell biological data overall should be improved. Not all figures have scale bars, some of the images could be better labeled with genotypes and arrows pointing out key features, and often not a sufficiently high enough level of magnification is used to interpret the images. Specific examples are provided in the Figure-by-Figure comments below.

We have corrected these issues as described below.

Figure 1

For all experiments it is not entirely clear exactly how many histone genes were in the cells that were analyzed; i.e. 12x or 24x, 100x or 200x? More clarity on this issue will help the reader think about how the data were normalized for presentation.

We have clarified this in the text on line 102; we are presenting the genomic numbers, not diploid cell numbers. So, the 12X genotype contains 12 histone copies-per-genome, 24 copies per cell.

Was Dm6 used for Figure 11, or was a single histone repeat unit that aggregates all the reads used? If the former, did the authors just choose a single HRU from the data in panel a, and is this representative? This question is important to interpret that shape of the cut and tag signal over the individual histone genes.

We used the dm6 assembly. This assembly contains 23 copies of three slightly different histone repeat units. The one instance we show in Figure contains all features of the histone repeat unit (the other copies are deleted for one or another small element). Thus it is representative of all potential histone repeat unit features.

Can the authors comment on why there is no (or very low) Mute signal on the H1 gene, especially considering the recent papers on CRAMP1 (which did come out after submission). PMIDs: 40516529, 40516528, 39829857.

We are happy to comment on these papers. Together with our work, they nicely corroborate the idea that histone H1 gene regulation is distinct from that of the other histone repeat unit genes. We now cite this work on lines 83-85.

Figure 2

More detail about the features of this table would be helpful. e.g. For each mark how were cutoffs for biological significance determined and why? What numbers in panel a are “important”? What is difference between grey font and black font?

We presented this data in a relatively raw form so that readers could assess for themselves without having to impose cutoffs, but that requires a fair amount of effort. We have improved this table by removing rows for modifications that are absent from the histone locus. The adjusted table replaces Figure 2a. Gray values were indicating that fold-change ratios from background-level modifications are noisy, but in the revised table this color was removed.

What does the green heat map represent?

This was log₂(CPM) for each modification across the Histone Repeat Unit in wildtype cells. We have removed this column for simplicity.

Figure 3

What is the exact genotype of the tissues. Most important, is “wildtype” a 200X or 100X because it’s a sibling of the 12XWT.

The strains are homozygous for their respective histone loci, and we have added this to the Figure legend (lines 697-700): we note in the legend that wildtype has 100 copies-per-genome, and the 12XWT strain has 12 copies-per-genome.

And how was the average summed fluorescence determined? Field of view (the entire disc or a portion) and how many discs?

We now more thoroughly describe this in the Methods. We imaged wildtype and 12XWT wing discs with a 20X lens and the same camera settings at an undersaturated exposure, then summed the fluorescence of the entire disc area (determined from a phase contrast image) for 5 wing discs of each genotype. We calculated the mean fluorescence of discs for each genotype, and then the ratio of these means to estimate the upregulation of the histone Dendra2 reporter.

Figure 4

The DAPI staining is strange; it doesn't appear if nuclei are being labeled.

DAPI staining of nuclei in the testis varies quite dramatically between the small nuclei of proliferative cells and the much larger decondensed nuclei of spermatocytes. We have added a comment to Figure legend (line 706) explaining this.

Carrying the dashed lines over to the color panels would be helpful.

Done.

Need to point to the sheath cells with an arrow and show a high mag view; they are nearly impossible to see as is.

We have added an arrowhead to point out these cells in Figure 4f.

A high mag image for panel h like in panels c, e, g would be helpful.

Added. We also included panels for the His3Dendra2 reporter in the 12XWT background (Figure 4j,k), and quantitation for all genotypes is presented in Figure 4l.

Panel g doesn't show a sheath cell as the legend indicates; those look like the gonial cells.

The legend was ambiguous; it describes panels f and g together, and sheath cells are present only in panel f; panel g is indeed only gonial cells. We have split up the legend for these panels for clarity.

The bottom line is that without more guidance the impact will be far lower for non Drosophila readers without this help.

We appreciate this concern, and expect that these suggested changes will improve the accessibility of our results.

Figure 4

They don't show any KD efficiencies for any of the target

We have added data as Supplementary Figure 2.

What is the asterisk in panel i?

Oops, this was an illustration error and is fixed.

What is the red/green cutoff based on? A quantifiable measurement or just inspection?

This was a visual screen with a digital microscope set to consistent exposures. Since in wildtype and most cases expression is at background levels (Figures 4d-g), visual inspection is quite sensitive to any increase in expression.

They don't formally show a control for the RNAi knockdowns

This data is now presented in Supplementary Figure 2.

Scale bars on all the figures and the actual scale is needed.

We have ensured that scale bars for all figures are present.

Figure 5

Discussed above.

The error in the legend that was confusing has been corrected.

Figure 6 and 7

There is no quantification of signals in any HLBs; measuring the intensity of MPM2 and S5p in the HLBs would increase the rigor of the conclusion that H4 KD increase S5p signal.

We agree and have added these measurements to Figure 7.

Figure 8

Panel of heat map is blurry and needs to be improved.

Fixed.

Reviewer #3 (Remarks to the Author):

The manuscript by Ahmad and colleagues uncovers a novel regulatory mechanism that couples DNA synthesis and histone deposition whereby cells prevent excess histone synthesis by histone H4 binding to the promoters of canonical histone genes. Using an elegant cellular system in fly, and molecular and cellular approaches, they demonstrate that histone H4 is specifically targeted to Histone Locus bodies (HLB), where it binds to promoters, and that the level of H4 negatively correlates with histone gene expression and HLB activity. Previously, they characterized the molecular requirements that define active histone unit repeats by comparing wild type genotype with a strain with a minimal (and highly active) set of histone copies. The text is clear, evidence is solid and, importantly, the conclusions are highly relevant, supporting a novel, simple and apparently conserved model to prevent the genotoxic effects of unbalanced histone synthesis.

Several points should be addressed before its publication.

1. An interesting aspect of the work is the conservation of the mechanism in human cells. Data clearly show that histone H4 binds to promoters of NPAT-transcriptionally activated histone genes. As indicated by the authors, this implicates histone H4 in the S phase regulation of canonical histone gene promoters in human cells. However, authors should demonstrate that partial histone H4 depletion also causes histone derepression in human cells. The fact that both H4 and NPAT coincides at promoters suggests that the expression level is modulated and could be potentially increased (after H4 reduction) or decreased (after NPAT reduction).

We have revised the Discussion to include a recent preprint that cites our preprint and acknowledges our discussions with them as inspiration for their work (<https://www.biorxiv.org/content/10.1101/2025.07.03.663002v1>) (lines 339-359). This group was studying the arginine symmetrical di-methyltransferase, PRMT5, one of the most promising epigenetic anti-cancer drug targets, but its function has been mysterious. They went on to confirm our results on HLB localization of histone H4, and to demonstrate that PRMT5-mediated histone H4 arginine-3 di-methylation disrupts this localization. The importance of this pathway is highlighted by their demonstration that the primary effect of anti-cancer PRMT5 inhibitors is to down-regulate histone gene expression, and not through affecting mRNA splicing has been previously believed. We believe that our two papers together stimulate substantial interest in this area.

2. Figures 4j-m and 7c-f: Even though the effect displayed in the representative images are clear, a quantification of the effects would be advisable.

We now present these quantitations in Figures 4 and 7.

3. Some controls should be included in the following panels:

- 4h: RFP and merge (they are modified testis and gonial cells should be shown)

We have added a high magnification image of gonial cells from this genotype. Adding RFP requires isolating a recombinant genotype with four elements, which we have not recovered; instead we identify gonial cells by phase imaging morphology. We now mention this in the figure legend.

- 5h or 5i: Histone H3 control; especially because a specific increase in H3K27ac is observed. Same for Figure 8 (either a or b), especially taking into account that authors try to demonstrate that the mechanism is conserved.

We don't have a suitable antibody that detects all histone H3 (H3.2 and H3.3 in Drosophila, H3.1, H3.2, H3.3 in humans). For Drosophila we address this with histone GFP fusion proteins in Figure 5a,b.

Minor points:

- Fig 2: Please list 2a and 2b in the same order for clarity with the text.

We reordered the tracks in Figure 2b to match the order of Figure 2a.

- Lines 143/144: the distribution of epigenetic marks does not allow to rule out the possibility that, rather than a mixture of active and inactive copies, as in the ribosomal locus, histones copies in the array are all partially expressed.

We have included this possibility in the text (line 109-112).

- Fig 7 text: c,e is c,d; and d,f is e,f.

Fixed.

Reviewer #1 (Remarks to the Author):

The authors addressed all the points I raised and I am in full support for this article to be published in NSMB. The complementary manuscript on H4 arginine-3 di-methylation further validates the importance of the current manuscript.

Reviewer #2 (Remarks to the Author):

The authors made an excellent effort to address all the reviewer comments, and the manuscript is clearer as a result. I also feel that the Discussion is improved and better tied to the results presented. I only have a few remaining thoughts and simple suggestions below.

Could also cite PMID: 40516529 and PMID: 21336627 on line 85.

We have added these references.

Could also cite PMID: 40397569 on line 94 and lines 249-253 where the results are similar to those in wing discs, and line 303.

We have added this reference to line 94 and to line 249.

The conclusion made in lines 111-112 was also made in PMID: 25669886, which could be cited here.

Done.

In Figure 5f you might show the pink EdU channel separately, as you did in panel e, to make it abundantly clear that that nucleus is not in S phase like the one in panel e.

We have added this separate panel to the Figure.

The knockdown of H4 in the HLB of testes in Supplemental Figure 2 is not convincingly shown (or very understandable), and there is no figure legend to help understand the data (e.g. what is the arrowhead pointing out?). A better presentation of these data is needed for them to be interpreted appropriately.

We added a zoomed inset to the figure to show the difference in staining of the HLB.

I still think the authors should use total histone gene count (i.e. 200x and 24x) per diploid genome rather than 100x and 12x haploid “copies/genome” number; it just seems more intuitive to me (this is a very minor point).

We now provide both per-genome numbers and per-cell numbers to make it clear to readers.

Reviewer #3 (Remarks to the Author):

All my concerns have been properly addressed, and the changes introduced in response to reviewers' critical points have successfully improved the manuscript, which provides a simple and attractive mechanism to understand how histone synthesis is coupled to histone demand.